# Rao-Blackwellised Reparameterisation Gradients

**Kevin H. Lam**
Department of Statistics
University of Oxford

**Thang D. Bui**
School of Computing
Australian National University

**George Deligiannidis**
Department of Statistics
University of Oxford

**Yee Whye Teh**
Department of Statistics
University of Oxford

{lam,deligian,y.w.teh}@stats.ox.ac.uk, thang.bui@anu.edu.au

## Abstract

Latent Gaussian variables have been popularised in probabilistic machine learning. In turn, gradient estimators are the machinery that facilitates gradient-based optimisation for models with latent Gaussian variables. The reparameterisation trick is often used as the default estimator as it is simple to implement and yields low-variance gradients for variational inference. In this work, we propose the R2-G2 estimator as the Rao-Blackwellisation of the reparameterisation gradient estimator. Interestingly, we show that the local reparameterisation gradient estimator for Bayesian MLPs is an instance of the R2-G2 estimator and Rao-Blackwellisation. This lets us extend benefits of Rao-Blackwellised gradients to a suite of probabilistic models. We show that initial training with R2-G2 consistently yields better performance in models with multiple applications of the reparameterisation trick.

## 1 Introduction

Latent random variables are ubiquitous in probabilistic machine learning (ML) as they enable us to embed prior assumptions into models, such as uncertainty in model parameters and low-dimensional latent representations of observed data structures. Thus, latent random variables appear in a range of modelling tasks, including variational inference, generative modelling, and density estimation. A wide variety of modern models with latent random variables, particularly those that parameterise a neural network (NN), are trained with gradient-based optimisation. This presents a non-trivial problem on how analytical derivatives can incorporate stochasticity, and has led to substantial interest in *gradient estimators*. These estimators enable gradient-based optimisation by computing approximations of gradients that are compatible with automatic differentiation software. For comprehensive reviews of gradient estimators with latent continuous and discrete random variables, we refer the reader to Mohamed et al. (2020) and Huijben et al. (2023) respectively.

We revisit the class of gradient estimators for latent Gaussian variables (Kingma and Welling, 2014; Rezende et al., 2014; Titsias and Lázaro-Gredilla, 2014), as Gaussian variables often serve as the default distribution for the noise or prior in probabilistic machine learning tasks. These estimators, known as pathwise estimators or the reparameterisation trick, have been popularised as they produce low-variance gradients for variational inference. In particular, it has been shown in Xu et al. (2019) that the reparameterisation gradient estimator has lower variance than a Rao-Blackwellised REINFORCE estimator (Williams, 1992) for variational inference. These estimators are also attractive as they are *single-sample* estimators since training models with them only requires sampling operations be done

once. In turn, this also limits costly function evaluations needed for training as models grow larger in the era of deep learning.

In this work, we present the R2-G2 estimator as an extension of the reparameterisation gradient estimator formed by conditioning on pre-activations in backpropagation. Notably, we show that the local reparameterisation gradient estimator proposed in Kingma et al. (2015) is an instance of the R2-G2 estimator for linear layers in a Bayesian MLP with independent weights and establish that it is equivalent to a Rao-Blackwellised reparameterisation gradient estimator.

Our main contributions are as follows:

- We present R2-G2, as a novel, general-purpose and single-sample gradient estimator for latent Gaussian variables as the Rao-Blackwellised reparameterisation gradient estimator;

- We show that the local reparameterisation gradient estimator is an instance of the R2-G2 estimator and enjoys variance reduction in gradients from Rao-Blackwellisation;

- We empirically show that initial training with the R2-G2 estimator consistently yields higher likelihoods on Bayesian Neural Networks and higher ELBOs on hierarchical Variational Autoencoders than the reparameterisation gradient estimator.

## 2   Problem setting

Let $\mathbf{v} \in \mathbb{R}^n$ be a vector of continuous random variables where we have $\mathbf{v} \sim q_{\boldsymbol{\theta}}$ for some probability distribution $q$ parameterised by a vector $\boldsymbol{\theta}$. Suppose we are provided a continuously differentiable loss function $\ell : \mathbb{R}^n \to \mathbb{R}$ that depends on the random variables $\mathbf{v}$. To enable gradient-based training of a parametric model $f$, our goal is to compute the gradient of the expected loss

$$\nabla_{\boldsymbol{\theta}} \mathbb{E}_{q_{\boldsymbol{\theta}}}[\ell_{\mathcal{D}, f}(\mathbf{v})] \tag{1}$$

where $\mathcal{D}$ is a dataset and $f$ is a parametric function optimised by the loss such as a neural network (NN), with both affecting the evaluation of $\ell$ (i.e. $\ell_{\mathcal{D}, f}$). We note that $\mathcal{D}$ is described generally here to encompass both supervised and unsupervised problems. For ease of notation in the remainder of this work, we will shorten our notation of the loss from $\ell_{\mathcal{D}, f}(\mathbf{v}, \boldsymbol{\theta})$ to $\ell$ where the context is clear with its dependence on $\mathbf{v}, \boldsymbol{\theta}, \mathcal{D}$, and $f$ implied. Popular choices of $\ell$ aim to maximise the log-likelihood or the evidence lower bound (ELBO) for the dataset $\mathcal{D}$. The former involves setting $\ell$ as the negative log-likelihood $\ell_{NLL}$. The latter sets $\ell$ as

$$\ell_{ELBO}(\mathbf{v}) = \ell_{NLL}(\mathcal{D}|\mathbf{v}) + \log\left(\frac{q_{\boldsymbol{\theta}}(\mathbf{v}|\mathcal{D})}{p(\mathbf{v})}\right)$$

where $p$ is a prior distribution of $\mathbf{v}$ and setting $q_{\boldsymbol{\theta}}$ as the posterior of $\mathbf{v}$ or its approximation, making Equation 1 equivalent to maximising the ELBO. Moreover, it is equivalent to variational Bayesian inference when the form of $q_{\boldsymbol{\theta}}$ is restricted as it becomes an approximation of the posterior.

In practice, we use a Monte Carlo approximation $\mathbb{E}_{q_{\boldsymbol{\theta}}}[\ell(\mathbf{v}, \boldsymbol{\theta})] \approx \frac{1}{M}\sum_{i=1}^{M} \ell(\mathbf{v}^{(i)}, \boldsymbol{\theta})$ to write

$$\nabla_{\boldsymbol{\theta}} \mathbb{E}_{q_{\boldsymbol{\theta}}}[\ell(\mathbf{v})] \approx \frac{1}{M}\sum_{i=1}^{M} \nabla_{\boldsymbol{\theta}} \ell(\mathbf{v}^{(i)})$$

where $\mathbf{v}^{(1)}, \ldots, \mathbf{v}^{(M)} \sim q_{\boldsymbol{\theta}}$. However, this would require $M$ evaluations of the loss $\ell$ which is not desirable when the underlying model $f$ is large or costly to evaluate. In this work, we focus on the setting where $\mathbf{v}$ are *independent* Gaussian random variables with the aim to derive a *single-sample* gradient estimator (i.e. $M = 1$) that is unbiased and enjoys reduced variance.

## 3   Related work

In this section, we revisit the single-sample gradient estimators compatible with Gaussian random variables, namely REINFORCE, reparameterisation trick and local reparameterisation trick.

**REINFORCE** The REINFORCE gradient estimator, also known as the *score function estimator*, is proposed by using the log-derivative trick: $\nabla_{\boldsymbol{\theta}} \log q_{\boldsymbol{\theta}}(\mathbf{v}) = \frac{\nabla_{\boldsymbol{\theta}} q_{\boldsymbol{\theta}}(\mathbf{v})}{q_{\boldsymbol{\theta}}(\mathbf{v})}$ which comes as a result of the chain rule (Glynn, 1990; Williams, 1992). Formally, the partial derivatives of parameters $\boldsymbol{\theta}$ are given by the REINFORCE estimator

$$\widehat{\nabla_{\boldsymbol{\theta}}\ell}^{SCORE} = \ell \cdot (\nabla_{\boldsymbol{\theta}} \log q_{\boldsymbol{\theta}})$$

It is an unbiased estimator of $\nabla_{\boldsymbol{\theta}} \mathbb{E}_{q_{\boldsymbol{\theta}}}[\ell]$, and only requires that one is able to evaluate $q_{\boldsymbol{\theta}}$ and sample from it. The latter requirement is not restrictive and easily achieved in most settings since the form of $q_{\boldsymbol{\theta}}$ is often assumed as part of model specification such as variational inference. Despite these desirable properties, it is well-known that the REINFORCE estimator suffers from high variance (Greensmith et al., 2001; Xu et al., 2019).

**Reparameterisation Trick** The reparameterisation trick is compatible with random variables that have a location-scale parameterisation or have tractable inverse cumulative distribution functions (CDFs)(Price, 1958; Glasserman, 2003; Kingma and Welling, 2014; Rezende et al., 2014; Titsias and Lázaro-Gredilla, 2014; Figurnov et al., 2018; Jankowiak and Obermeyer, 2018). For Gaussian random variables, they are reparameterisable using a location-scale transformation which lets us write

$$\mathbf{v} \sim \mathcal{N}(\boldsymbol{\mu}, \boldsymbol{\Sigma}) \stackrel{d}{=} g(\boldsymbol{\epsilon}, \boldsymbol{\mu}, \mathbf{V}) = \boldsymbol{\mu} + \mathbf{V}\boldsymbol{\epsilon}$$

where $\mathbf{V} \in \mathbb{R}^{n \times n}$, $\boldsymbol{\Sigma} = \mathbf{V}\mathbf{V}^{\top}$ and $\boldsymbol{\epsilon} \sim q_0(\boldsymbol{\epsilon}) = \mathcal{N}(\mathbf{0}, \mathbf{I})$. The latter expression for $\mathbf{v}$ is compatible with automatic differentiation and enables gradient-based updates to $\boldsymbol{\mu}$ and $\mathbf{V}$. Typically, $\mathbf{V}$ is a diagonal matrix parameterised by a vector of variances $\boldsymbol{\tau} = (\sigma_1^2, \ldots, \sigma_n^2)$. That is, $\mathbf{V} = \boldsymbol{\Sigma}^{\frac{1}{2}}$ with $\boldsymbol{\Sigma} = \texttt{diag}(\boldsymbol{\tau})$, so we have $\boldsymbol{\theta} = \{\boldsymbol{\mu}, \boldsymbol{\tau}\}$ as our training parameters. Formally, we can write $\ell(g(\boldsymbol{\epsilon}, \boldsymbol{\theta}))$ and apply the chain rule to yield the reparameterisation gradient estimator

$$\widehat{\nabla_{\boldsymbol{\theta}}\ell}^{RT} = \left( J_\ell(\mathbf{v}) \cdot \left[ \ \mathbf{I}_n \ \middle| \ \tfrac{1}{2}\boldsymbol{\Sigma}^{-\frac{1}{2}} \odot (\mathbf{1}_n \boldsymbol{\epsilon}^{\top}) \ \right] \right)^{\top} \tag{2}$$

where $J_\ell$ is the Jacobian of $\ell$ with respect to $\mathbf{v}$, and $[A|B]$ denotes a partitioned matrix with block matrices $A$ and $B$. The terms in the matrix are derived from the Jacobians $J_g(\boldsymbol{\mu}) = \mathbf{I}_n$ and $J_g(\boldsymbol{\tau}) = \frac{1}{2}\boldsymbol{\Sigma}^{-\frac{1}{2}} \odot (\mathbf{1}_n \boldsymbol{\epsilon}^{\top})$. Equation 2 is an unbiased estimator of $\nabla_{\boldsymbol{\theta}} \mathbb{E}_{q_{\boldsymbol{\theta}}}[\ell]$ due to the equivalence in distribution achieved by the reparameterisation trick. In the context of probabilistic modelling, the reparameterisation trick is often applied to individual scalar inputs of a decoder within a variational autoencoder (VAE) or individual weights in a Bayesian neural network (BNN). The latter may also be referred to as the *global* reparameterisation trick.

**Local Reparameterisation Trick** The local reparameterisation trick was proposed for NNs where the *weights of linear layers* are independent Gaussian random variables (Kingma et al., 2015). Given an input $\mathbf{x} \in \mathbb{R}^n$, the pre-activations of these linear layers are given by

$$\mathbf{z} = \begin{bmatrix} \mathbf{x}^{\top} g(\boldsymbol{\epsilon}^{(1)}, \boldsymbol{\theta}^{(1)}) \\ \vdots \\ \mathbf{x}^{\top} g(\boldsymbol{\epsilon}^{(m)}, \boldsymbol{\theta}^{(m)}) \end{bmatrix} \in \mathbb{R}^m$$

where $\boldsymbol{\epsilon}^{(i)}$ and $\boldsymbol{\epsilon}^{(j)}$ are independent for all $i \neq j$, and they admit a factorised Gaussian distribution $\tilde{q}_{\mathbf{z}} = \prod_{i=1}^{m} \tilde{q}_{z_i}$ where each pre-activation $z_i \sim \tilde{q}_{z_i}$. Instead of sampling $g$ and computing $\mathbf{x}^{\top} g$, it is more efficient to *locally* apply the reparameterisation trick to directly sample scalar pre-activations

$$z_i \sim \mathcal{N}\left( \sum_{j=1}^{n} x_j \mu_j^{(i)}, \sum_{j=1}^{n} x_j^2 \left(\sigma_j^{(i)}\right)^2 \right) \stackrel{d}{=} \sum_{j=1}^{n} x_j \mu_j^{(i)} + \left( \sum_{j=1}^{n} x_j^2 \left(\sigma_j^{(i)}\right)^2 \right)^{\frac{1}{2}} \xi_i \tag{3}$$

where $\boldsymbol{\mu}^{(i)}$ and $\boldsymbol{\tau}^{(i)} = \left( \left(\sigma_1^{(i)}\right)^2, \ldots, \left(\sigma_n^{(i)}\right)^2 \right)$ are the *global* mean and variance parameters of the Gaussian variables used to compute $z_i$ respectively, $\boldsymbol{\theta}^{(i)} = \{\boldsymbol{\mu}^{(i)}, \boldsymbol{\tau}^{(i)}\}$, and $\xi_i \sim q_0^{(i)} = \mathcal{N}(0, 1)$. The latter expression in Equation 3 is known as the local reparameterisation trick as it changes the parameterisation of each $z_i$ as a scalar function of $\boldsymbol{\epsilon}^{(i)}$ and $\boldsymbol{\theta}^{(i)}$. Formally, we can write $\ell(g(\boldsymbol{\epsilon}, \boldsymbol{\theta})) = \tilde{\ell}\left( z_1\left(\xi_1, \boldsymbol{\theta}^{(1)}\right), \ldots, z_m\left(\xi_m, \boldsymbol{\theta}^{(m)}\right) \right)$ where $\tilde{\ell} : \mathbb{R}^m \to \mathbb{R}$ denotes functions applied to $\mathbf{z}$ to compute

Table 1: Examples of linear maps applied to Gaussian variables in probabilistic neural networks.

| Application | $\mathbf{W}$ | $\mathbf{V}\boldsymbol{\epsilon}$ |
|---|---|---|
| BNN | Outputs of a previous hidden layer | Gaussian weights in a linear layer |
| BNN | Vectorised patches of an image | Gaussian weights in a convolutional layer |
| VAE | Linear layer after reparameterisation | Gaussian latent variables |

the loss $\ell$ (i.e. non-linearities and upper NN layers), and apply the chain rule to yield the local reparameterisation gradient estimator for each $\boldsymbol{\theta}^{(i)}$

$$\widehat{\nabla_{\boldsymbol{\theta}^{(i)}}\ell}^{LRT} = \left( \frac{\partial \tilde{\ell}}{\partial z_i} \cdot \left[ \ \mathbf{x}^\top \ \middle| \ \tfrac{1}{2} \left( \textstyle\sum_{j=1}^n x_j^2 \left( \sigma_j^{(i)} \right)^2 \right)^{-\frac{1}{2}} \xi_i \left( \mathbf{x} \odot \mathbf{x} \right)^\top \ \right] \right)^\top \tag{4}$$

where $\odot$ denotes element-wise product of matrices. The terms in the matrix are derived from the Jacobians $J_{z_i}(\boldsymbol{\mu}^{(i)}) = \mathbf{x}^\top$ and $J_{z_i}(\boldsymbol{\tau}^{(i)}) = \tfrac{1}{2} \left( \sum_{j=1}^n x_j^2 \left( \sigma_j^{(i)} \right)^2 \right)^{-\frac{1}{2}} \xi_i \left( \mathbf{x} \odot \mathbf{x} \right)^\top$. Equation 4 is an unbiased estimator of $\nabla_{\boldsymbol{\theta}} \mathbb{E}_{q_{\boldsymbol{\theta}^{(i)}}}[\ell]$ due to the equivalence in distribution achieved by the local reparameterisation trick. A notable application of the local reparameterisation trick is when performing mean-field variational inference within the linear layer of a BNN. It has been empirically observed that Equation 4 has lower variance than Equation 2 (Kingma et al., 2015). In Theorem 4.3, we show that the local reparameterisation gradient estimator is equivalent to applying Rao-Blackwellisation to the reparameterisation gradient estimator and thereby enjoys variance reduction benefits of Rao-Blackwellisation.

## 4 R2-G2 Gradient Estimator

The reparameterisation trick can be seen as a procedure that couples the specification of a probabilistic model with an expression for gradients. In turn, assumptions on the former would restrict the applicability of the latter. This highlights the disadvantage of the local reparameterisation trick: the probabilistic model and gradients induced by the local reparameterisation trick is not suitable for general settings where the pre-activations $\mathbf{z}$ do not admit a factorised Gaussian distribution (i.e. the covariance matrix of $\mathbf{z}$ is not diagonal). On the other hand, the local reparameterisation trick has been empirically shown to enjoy lower variance of gradients than the global reparameterisation trick.

To get the best of both worlds, we seek a reparameterisation gradient estimator that is general-purpose and enjoys reduced variance. To this end, we present our contribution: the **R**ao-Blackwellised **R**eparameterisation **G**radient Estimator for **G**aussian random variables, coined the R2-G2 estimator, as the Rao-Blackwellisation of the reparameterisation gradient estimator by conditioning on the realisation of multivariate Gaussian vectors resultant from linear transformations of Gaussian vectors. We first describe the Rao-Blackwellisation of the reparameterisation gradient estimator. We then provide the analytical form of the R2-G2 estimator and a summary of key properties of the R2-G2 estimator and its connection to existing methods. In particular, we show that the local reparameterisation gradient estimator is an instance of the R2-G2 estimator. We then conclude with a practical implementation to bypass costly matrix inversion operations by reformulating matrix-vector products as the solution to a quadratic optimisation problem. We defer all proofs to Appendix B.

### 4.1 Rao-Blackwellisation of the Reparameterisation Gradient Estimator

The idea behind our Rao-Blackwellisation scheme is to condition on linear transformations of Gaussian variables since conditional Gaussian distributions can be described analytically. This is further motivated by the observation that we can decompose loss evaluations in NNs as

$$\ell(g(\boldsymbol{\epsilon}, \boldsymbol{\theta})) = (\tilde{\ell} \circ \mathbf{W})(g(\boldsymbol{\epsilon}, \boldsymbol{\theta})) \tag{5}$$

where $\mathbf{W} : \mathbb{R}^n \to \mathbb{R}^m$ is a linear map and $\mathbf{W} \cdot g(\boldsymbol{\epsilon}, \boldsymbol{\theta})$ are pre-activations. This generalises the idea of the local reparameterisation trick where $\mathbf{W}$ is a row vector (i.e. $\mathbf{W} = \mathbf{x}^\top$) and inputs of $\tilde{\ell}$ are $m$ scalar pre-activations. In deep learning, $\mathbf{W}$ frequently appears as hidden layers within NNs. A list of

common linear transformations in probabilistic NNs is given in Table 1. Applying the chain rule to Equation 5 gives an alternative expression of the reparameterisation gradient estimator

$$\widehat{\nabla_{\boldsymbol{\theta}}\ell}^{RT} = \left( J_{\tilde{\ell}}(\mathbf{W}\cdot g)\cdot\mathbf{W}\cdot\left[\ \mathbf{I}_n\ \middle|\ \tfrac{1}{2}\boldsymbol{\Sigma}^{-\frac{1}{2}}\odot(\mathbf{1}_n\boldsymbol{\epsilon}^\top)\ \right]\right)^\top \tag{6}$$

where $J_{\tilde{\ell}}$ is the Jacobian of $\tilde{\ell}$. With Equation 6 in hand, we now present the R2-G2 estimator[1].

**Definition 4.1 (R2-G2)** *The R2-G2 gradient estimator is given by*

$$\widehat{\nabla_{\boldsymbol{\theta}}\ell}^{R2\text{-}G2} = \mathbb{E}_{\tilde{q}_0}\left[\widehat{\nabla_{\boldsymbol{\theta}}\ell(\boldsymbol{\epsilon})}^{RT}\right] = \left( J_{\tilde{\ell}}(\mathbf{W}\cdot g)\cdot\mathbf{W}\cdot\mathbb{E}_{\tilde{q}_0}\left[\ \mathbf{I}_n\ \middle|\ \tfrac{1}{2}\boldsymbol{\Sigma}^{-\frac{1}{2}}\odot(\mathbf{1}_n\boldsymbol{\epsilon}^\top)\ \right]\right)^\top \tag{7}$$

*where $\boldsymbol{\epsilon}|\mathbf{W}\cdot g = \mathbf{z} \sim \tilde{q}_0$ with mean $\boldsymbol{\epsilon}^* = \mathbf{A}^\top\left(\mathbf{A}\mathbf{A}^\top\right)^\dagger(\mathbf{z}-\mathbf{W}\boldsymbol{\mu}) = \mathbf{A}^\top\left(\mathbf{A}\mathbf{A}^\top\right)^\dagger\mathbf{A}\boldsymbol{\epsilon}$ with $\mathbf{A} = \mathbf{W}\mathbf{V}$. By linearity of expectations, it also admits a closed-form expression*

$$\widehat{\nabla_{\boldsymbol{\theta}}\ell}^{R2\text{-}G2} = \left( J_{\tilde{\ell}}(\mathbf{W}\cdot g)\cdot\mathbf{W}\cdot\left[\ \mathbf{I}_n\ \middle|\ \tfrac{1}{2}\boldsymbol{\Sigma}^{-\frac{1}{2}}\odot(\mathbf{1}_n(\boldsymbol{\epsilon}^*)^\top)\ \right]\right)^\top. \tag{8}$$

The R2-G2 estimator is a single-sample gradient estimator as it is a conditional expectation of the reparameterisation gradient estimator formed by conditioning on a single sample of the vector of pre-activations $\mathbf{z}$, making it the Rao-Blackwellisation (Blackwell, 1947; Rao et al., 1992) of the reparameterisation gradient estimator from Equation 6. The law of total variance lets us deduce that the R2-G2 estimator has lower variance since it swaps the random matrix in Equation 6 with its conditional expectation. We conclude our description of the R2-G2 estimator with its key properties, namely unbiasedness and enjoying lower variance than the reparameterisation gradient estimator.

**Proposition 4.2** *Denote $\mathbf{z} \sim q_{\mathbf{z}} = \mathcal{N}\left(\mathbf{W}\cdot\boldsymbol{\mu}, \mathbf{A}\mathbf{A}^\top\right)$. Then we have*

$$\mathbb{E}_{q_{\mathbf{z}}}\left[\widehat{\nabla_{\boldsymbol{\theta}}\ell}^{R2\text{-}G2}\right] = \nabla_{\boldsymbol{\theta}}\mathbb{E}_{q_{\boldsymbol{\theta}}}[\ell]$$

*and*

$$\mathbb{E}_{q_{\mathbf{z}}}\left[\left\|\widehat{\nabla_{\boldsymbol{\theta}}\ell}^{R2\text{-}G2} - \nabla_{\boldsymbol{\theta}}\mathbb{E}_{q_{\boldsymbol{\theta}}}[\ell]\right\|^2\right] \leq \mathbb{E}_{q_0}\left[\left\|\widehat{\nabla_{\boldsymbol{\theta}}\ell}^{RT} - \nabla_{\boldsymbol{\theta}}\mathbb{E}_{q_{\boldsymbol{\theta}}}[\ell]\right\|^2\right].$$

### 4.2 Computation of Rao-Blackwellisation scheme as a least-squares problem

The R2-G2 estimator exploits the analytical form of the conditional Gaussian distribution. This is done by either sampling from the conditional Gaussian distribution $\tilde{q}_0$ within Equation 7 or directly using the mean of the conditional Gaussian distribution $\boldsymbol{\epsilon}^*$ in place of $\boldsymbol{\epsilon}$ within Equation 8. Equation 7 requires computing the Cholesky factor of the covariance matrix of $\tilde{q}_0$, which incurs a $\mathcal{O}(m^3)$ computational cost and a $\mathcal{O}(m^2)$ storage cost. The latter makes it impractical for training NNs as each gradient descent step would require instantiating the matrix $\mathbf{A}\mathbf{A}^\top$. This makes Equation 8 more desirable as long as $\boldsymbol{\epsilon}^*$ is computed and stored efficiently. To do so, we consider the linear system

$$\mathbf{A}\mathbf{A}^\top\boldsymbol{\beta} = \mathbf{A}\boldsymbol{\epsilon}. \tag{9}$$

It can be verified that any solution $\boldsymbol{\beta}^*$ to Equation 9 satisfies $\boldsymbol{\epsilon}^* = \mathbf{A}^\top\boldsymbol{\beta}^*$ (see Appendix D).

**Least-squares characterisation of Rao-Blackwellisation** Observe that Equation 9 is a normal equation that describes the first-order optimality condition of the quadratic optimisation problem

$$\underset{\boldsymbol{\beta}\in\mathbb{R}^m}{\text{minimise}}\quad \frac{1}{2}\|\boldsymbol{\epsilon}-\mathbf{A}^\top\boldsymbol{\beta}\|_2^2 \Leftrightarrow \underset{\boldsymbol{\beta}\in\mathbb{R}^m}{\text{maximise}}\quad \exp\left(-\frac{1}{2}\|\boldsymbol{\epsilon}-\mathbf{A}^\top\boldsymbol{\beta}\|_2^2\right).$$

We can interpret computing a solution $\boldsymbol{\beta}^*$ as fitting a single-sample multivariate linear model $\boldsymbol{\epsilon}|\mathbf{A}$

$$\boldsymbol{\epsilon} = \mathbf{A}^\top\boldsymbol{\beta} + \boldsymbol{\delta}$$

---

[1]Equations 7 and 8 in Definition 4.1 only hold when $\mathbf{v}$ are independent Gaussian variables. A more general expression can be derived when independence does not hold.

where $\boldsymbol{\beta} \in \mathbb{R}^m$ and $\boldsymbol{\delta} \sim \mathcal{N}(\mathbf{0}_n, \mathbf{I}_n)$. Under this linear model, the goal is to solve for $\boldsymbol{\beta}^*$ by using covariance parameters $\mathbf{A}$ as *predictors* of the noise $\boldsymbol{\epsilon}$. With $\boldsymbol{\beta}^*$ computed, we can then compute

$$\boldsymbol{\epsilon}^* = \mathbf{A}^\top \boldsymbol{\beta}^* = \mathbf{A}^\top \left(\mathbf{A}\mathbf{A}^\top\right)^\dagger \mathbf{A}\boldsymbol{\epsilon} \tag{10}$$

which corresponds to the expression of the mean of a conditional Gaussian distribution (see Lemma A.1). In other words, computing the R2-G2 estimator is inherently solving a least squares problem and applying a linear transformation. Intuitively, the former is the mechanism giving variance reduction. In the language of linear models, the observed noise $\boldsymbol{\epsilon}$ is projected to the *fitted* noise $\boldsymbol{\epsilon}^*$ which has minimal Euclidean norm.

**Iterative solver** By defining functions that compute matrix-vector products with the matrix $\mathbf{A}\mathbf{A}^\top$, we can use the conjugate gradient algorithm to solve Equation 9 for $\boldsymbol{\beta}^*$. The conjugate gradient algorithm is an iterative method that terminates in at most $\mathtt{rank}(\mathbf{A}) \leq m$ iterations (Kaasschieter, 1988; Nocedal and Wright, 1999; Hayami, 2018), and only re-

---
**Algorithm 1** Forward Pass with R2-G2 Gradients.

---
**Input:** matrix $\mathbf{A}$, noise vector $\boldsymbol{\epsilon}$.
Compute $\mathbf{z} = \mathbf{A}\boldsymbol{\epsilon}$.
Compute $\boldsymbol{\beta}^* = \mathtt{conjugate\_gradient}(\mathbf{A}, \mathbf{z})$.
Compute $\boldsymbol{\epsilon}^* = \mathbf{A}^\top \boldsymbol{\beta}^*$.
Compute $\mathbf{z}^* = \mathbf{A}\boldsymbol{\epsilon}^*$.
**Output:** $\mathtt{stop\_gradient}(\mathbf{z} - \mathbf{z}^*) + \mathbf{z}^*$.

---

quires storing the updated solution at each iteration. While the former implies that the worst-case computational costs remains at $\mathcal{O}(m^3)$, the latter implies the storage cost is reduced to $\mathcal{O}(m)$ and makes it practical to compute $\boldsymbol{\beta}^*$. In our setting, the worst-case computational costs of $\mathcal{O}(m^3)$ can be reduced since the structure of $\mathbf{A}$ is known (see Appendix F). With $\boldsymbol{\beta}^*$ in hand, we can compute $\boldsymbol{\epsilon}^*$ and modify forward computations to use the R2-G2 gradient estimator for backpropagation[2], by implementing Algorithm 1 in deep learning frameworks such as PyTorch (Paszke et al., 2019).

### 4.3 Connections to related work

**Local Reparameterisation Gradient Estimator** By setting $\mathbf{V}^{(i)} = \left(\mathtt{diag}(\boldsymbol{\tau}^{(i)})\right)^{\frac{1}{2}}$, $\mathbf{W} = \mathbf{x}^\top$ and the conditional distribution $\boldsymbol{\epsilon}^{(i)}|z_i \sim \tilde{q}_0^{(i)}$ for $i = 1, \dots, m$ within Definition 4.1, we are able to show that the local reparameterisation gradient estimator is equivalent to the R2-G2 estimator. Theorem 4.3 presents this equivalence and formalises the empirical variance reduction of the local reparameterisation trick observed in the experiments of Kingma et al. (2015), as an instance of Rao-Blackwellisation and the R2-G2 estimator. Formally, it shows the local reparameterisation trick is equivalent to using pre-activation samples for forward computations and a Rao-Blackwellised *global* reparameterisation gradient estimator to update parameters in linear layers.

**Theorem 4.3** *Suppose we have a BNN linear layer where weights are independent Gaussian random variables. That is,* $\boldsymbol{\theta}^{(i)} = \left\{\boldsymbol{\mu}^{(i)}, \boldsymbol{\tau}^{(i)}\right\}$ *where* $\boldsymbol{\tau}^{(i)} = \left(\left(\sigma_1^{(i)}\right)^2, \dots, \left(\sigma_n^{(i)}\right)^2\right)$ *for* $i = 1, \dots, m$. *Then for each* $i = 1, \dots, m$, *we have* $\widehat{\nabla_{\boldsymbol{\theta}^{(i)}}\ell}^{R2\text{-}G2} = \mathbb{E}_{\tilde{q}_0^{(i)}}\left[\widehat{\nabla_{\boldsymbol{\theta}^{(i)}}\ell}^{RT}\right] \stackrel{d}{=} \widehat{\nabla_{\boldsymbol{\theta}^{(i)}}\ell}^{LRT}$ *and*

$$\mathbb{E}_{\tilde{q}_{z_i}}\left[\left\|\widehat{\nabla_{\boldsymbol{\theta}^{(i)}}\ell}^{LRT} - \nabla_{\boldsymbol{\theta}^{(i)}}\mathbb{E}_{q_{\boldsymbol{\theta}^{(i)}}}[\ell]\right\|^2\right] \leq \mathbb{E}_{q_0^{(i)}}\left[\left\|\widehat{\nabla_{\boldsymbol{\theta}^{(i)}}\ell}^{RT} - \nabla_{\boldsymbol{\theta}^{(i)}}\mathbb{E}_{q_{\boldsymbol{\theta}^{(i)}}}[\ell]\right\|^2\right].$$

As an analogy, we can view the way that R2-G2 generalises the local reparameterisation gradient estimator, in the way that square matrix inversion generalises scalar inversion. Recall that Equation 3 exploits the fact that the Gaussian pre-activations $\mathbf{z}$ of mean-field BNN linear layers have a diagonal covariance matrix *by design*. For forward computations, this means each $z_i$ can be efficiently sampled with the reparameterisation trick by using the square root of the variance $\sigma_{z_i}^2 = \sum_{j=1}^n x_j^2 \left(\sigma_j^{(i)}\right)^2$. The subtle benefit of reduced variance in gradients is due to the square root function having a derivative which matches its reciprocal (up to a scaling factor). This means that the variance $\sigma_{z_i}^2$ (i.e. a scalar) is *inverted* when using the local reparameterisation gradient estimator. The R2-G2 estimator naturally extends the inversion of scalar variances to square covariance matrices, by calling the conjugate gradient algorithm, thereby making it suitable for other probabilistic models.

---

[2]In Algorithm 1, computing $\mathbf{z}^*$ ensures automatic differentiation calculates Equation 8 for backpropagation, while the output $\mathtt{stop\_gradient}(\mathbf{z} - \mathbf{z}^*) + \mathbf{z}^*$ ensures the forward computation is still performed with $\mathbf{z}$.

Table 2: Log-likelihoods and classification accuracies (%) of BNNs using the R2-G2, Reparameterisation (RT) and Local Reparameterisation (LRT) estimators over 5 runs. Higher is better. Error bars denote $\pm 1.96$ standard errors ($\sigma/\sqrt{5}$) over 5 runs. See text for details.

| Dataset | Estimator | Log-likelihood | | Accuracy | |
|---------|-----------|-------|------|-------|------|
| | | Train | Test | Train | Test |
| MNIST | R2-G2 | $-\mathbf{3.00 \pm 0.00}$ | $-\mathbf{3.06 \pm 0.00}$ | $99.81 \pm 0.03$ | $\mathbf{98.00 \pm 0.08}$ |
| | LRT | $-\mathbf{3.00 \pm 0.00}$ | $-\mathbf{3.06 \pm 0.00}$ | $99.81 \pm 0.04$ | $97.99 \pm 0.07$ |
| | RT | $-\mathbf{3.00 \pm 0.00}$ | $-3.07 \pm 0.00$ | $\mathbf{99.85 \pm 0.04}$ | $\mathbf{98.00 \pm 0.11}$ |
| CIFAR-10 | R2-G2 | $-\mathbf{3.83 \pm 0.01}$ | $-\mathbf{3.87 \pm 0.01}$ | $\mathbf{70.08 \pm 0.43}$ | $67.97 \pm 0.66$ |
| | RT | $-3.84 \pm 0.01$ | $-3.88 \pm 0.02$ | $69.76 \pm 0.46$ | $\mathbf{67.98 \pm 0.88}$ |

**Variance reduction as Stochastic Linear Regression**  Variance reduction through stochastic linear regression has previously been explored in the context of variational inference Salimans and Knowles (2013, 2014); Salimans (2014). Specifically, variational inference is presented as fitting linear regression with the unnormalised log posterior as the dependent variable and the sufficient statistics of latent random variables as explanatory variables. The connection to variance reduction of gradients is then made by using *multiple* samples to construct control variates that correlate with the gradient of the KL-divergence. This differs from our work as we present variance reduction of gradients as Rao-Blackwellisation through the R2-G2 estimator as a *single-sample* gradient estimator, and explicate the connection to linear regression with the reparameterisation noise and covariance matrix parameters as the dependent and explanatory variables respectively.

## 5 Experiments

### 5.1 Protocol

The R2-G2 estimator can be readily applied to any existing application of the reparameterisation trick for Gaussian variables. While this may appear restrictive, we note that Gaussian distributions often serve as a default prior or noise distribution in probabilistic ML tasks. To motivate our experiments, we note that the R2-G2 and reparameterisation gradient estimators would yield the same model when a *large* number of gradient descent steps are taken, as the mean gradient of both estimators are the same. The focus of our experiments is during initial training, where only a *small* number of gradient descent steps are taken. Examples of scenarios where this setting can be beneficial include pre-training to discover a better initialisation for a model, fine-tuning a pre-trained model, or when the amount of training is limited by a computation budget.

We evaluate the benefits of initial training with the R2-G2 estimator for probabilistic models. We consider two standard tasks with variational Bayesian models that utilise a mean-field approximation of the posterior: image classification with BNNs and generative modelling with hierarchical VAEs. In our experiments, we provide numerical comparisons against the reparameterisation (RT) and local reparameterisation (LRT) gradient estimators, where permissible. We do not compare against the REINFORCE estimator as it is well-known that its high variance makes optimisation difficult. Unless stated otherwise, we compute pre-activations in the same way as the reparameterisation trick (i.e. sampling $g(\boldsymbol{\epsilon}, \boldsymbol{\theta})$ and computing $\mathbf{W} \cdot g$). We defer the full details of our experiments to Appendix E.

### 5.2 Image classification with Bayesian Neural Networks

We consider fully stochastic BNNs for image classification tasks on two standard benchmark datasets: MNIST (LeCun et al., 2010) and CIFAR-10 (Krizhevsky and Hinton, 2009). In this setting, the input data $\mathcal{D}$ is a dataset and the latent variables $\mathbf{v}$ are weights (i.e. $\mathbf{v}$ are global variables). To enable training with minibatches, we use the stochastic approximation of the variational lower bound

$$\log p(y|\mathbf{x}) > \mathbb{E}_{q_{\boldsymbol{\theta}}(\mathbf{v}|\mathcal{D})} \left[ \log \left( \frac{p(\mathbf{v})}{q_{\boldsymbol{\theta}}(\mathbf{v}|\mathcal{D})} \right) + \frac{N}{B} \sum_{i=1}^{B} \log p(y^{(i)}|\mathbf{x}^{(i)}, \mathbf{v}) \right]$$

where $N$ is the size of the dataset $\mathcal{D}$ and $\left\{(\mathbf{x}^{(i)}, y^{(i)})\right\}_{i=1}^{B} \subset \mathcal{D}$ is a minibatch of size $B < N$. We use the standard train and test splits of both datasets. See Table 2 for classification accuracies and log-likelihoods reported on the train and test sets of each dataset.

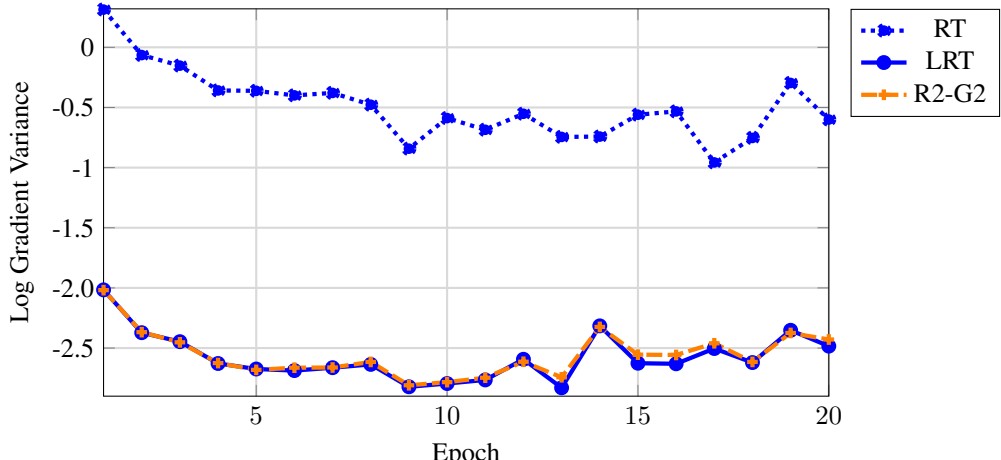

Figure 1: Log average gradient variance v.s. epoch for the top layer of a Bayesian MLP trained on *MNIST* over 5 runs. We compare the variance of gradients when training using the reparameterisation (RT), local reparameterisation (LRT) and R2-G2 estimators.

For MNIST, we used the two-layer multi-layer perceptron (MLP) architecture from Srivastava et al. (2014), which consists of two hidden layers of 1024 `ReLU` units. Each estimator is applied to all linear layers. For experiments with the R2-G2 estimator, we compute pre-activations in the same way as the local reparameterisation trick (i.e. sampling pre-activations directly). Our results illustrate that accuracies and log-likelihoods for the R2-G2 and LRT estimators match exactly. In Figures 1 and 3, the variance of gradients for the R2-G2 and LRT estimators also coincide and are much lower than those of the RT estimator. These observations empirically supports Theorem 4.3: the LRT estimator is an instance of the R2-G2 estimator and Rao-Blackwellisation. In other words, the local reparameterisation trick is equivalent to directly sampling pre-activations in forward computations and using a Rao-Blackwellised RT estimator in backpropagation.

For CIFAR-10, we used the VGG-11 architecture described in Simonyan and Zisserman (2015), where each estimator is applied to the last four convolutional layers. For the last four convolutional layers, we do not include the local reparameterisation trick as a benchmark since sampling *scalar* pre-activations directly would ignore dependencies induced by sharing weights in a convolutional layer (i.e. *additionally assumes* $\mathbf{A}\mathbf{A}^{\top}$ is diagonal). The RT estimator is applied to the first four convolutional layers. To limit the effect of gradient variances from linear layers on gradient variances of convolutional layers, we applied the LRT estimator to all linear layers.

Across both datasets, we found that initial training with the R2-G2 estimator yields higher log-likelihoods than the RT estimator while enjoying similar levels of accuracy. These results extend the benefits of training with Rao-Blackwellised gradients from Bayesian MLPs to Bayesian CNNs.

### 5.3 Generative modelling with Hierarchical Variational Autoencoders

We consider VAEs for generative modelling tasks on three standard benchmark datasets: MNIST (LeCun et al., 2010), Omniglot (Lake et al., 2015) and Fashion-MNIST (Xiao et al., 2017). We do not focus on comparisons of gradient variances due to architecture constraints of VAEs. Validating Proposition 4.2 requires independent and identically distributed (i.i.d.) samples of $\mathbf{z} \sim \tilde{q}_{\mathbf{z}}$ which requires computing a Cholesky factor in each iteration. This is computationally expensive and not representative of computations in VAEs. In practice, we compute $\mathbf{z} = \mathbf{W} \cdot g(\boldsymbol{\epsilon}, \boldsymbol{\theta})$ where $\mathbf{W}$ is the linear layer following reparameterisation, which does not enable sampling the full support of $\tilde{q}_{\mathbf{z}}$.

We found that training one-layer VAEs with the R2-G2 estimator did not guarantee performance gains (see Appendix H), and surmise that reducing gradient variance hinders the learning of stable

Table 3: Test variational lower bounds for hierarchical VAEs using the R2-G2 and Reparameterisation (RT) estimators. Higher is better. Error bars denote $\pm 1.96$ standard errors ($\sigma/\sqrt{5}$) over 5 runs. See text for details.

| # VAE Layers | Estimator | MNIST | Omniglot | Fashion-MNIST |
|---|---|---|---|---|
| 2 | R2-G2 | $\mathbf{-106.85 \pm 5.00}$ | $\mathbf{-129.80 \pm 0.74}$ | $\mathbf{-240.23 \pm 0.64}$ |
| | RT | $-107.64 \pm 8.46$ | $-131.48 \pm 1.66$ | $-240.59 \pm 0.93$ |
| 3 | R2-G2 | $\mathbf{-102.45 \pm 3.73}$ | $\mathbf{-134.95 \pm 2.26}$ | $\mathbf{-240.15 \pm 0.86}$ |
| | RT | $-111.50 \pm 5.40$ | $-136.12 \pm 2.70$ | $-240.89 \pm 0.97$ |

representations in this setting as only training of the encoder is affected by the R2-G2 estimator. An investigation of this phenomena is beyond the scope of this paper and we leave this for future work.

We present the results of our experiments for two-layer and three-layer VAEs (i.e. hierarchical VAEs). In this setting, the input data $\mathcal{D}$ is an observation and the latent variables $\mathbf{v}$ is its latent representation (i.e. $\mathbf{v}$ are local variables). The objective is to maximise the variational lower bound on the log-likelihood

$$\log p(\mathbf{x}) > \mathbb{E}_{q_{\boldsymbol{\theta}}(\mathbf{v}^{(1)}|\mathcal{D}),\ldots,q_{\boldsymbol{\theta}}(\mathbf{v}^{(K)}|\mathcal{D})} \left[ \log \left( \frac{1}{K} \sum_{i=1}^{K} \frac{p(\mathcal{D}, \mathbf{v}^{(i)})}{q_{\boldsymbol{\theta}}(\mathbf{v}^{(i)}|\mathcal{D})} \right) \right]$$

where $\mathcal{D}$ denotes input data, and $\{\mathbf{v}^{(i)}\}_{i=1}^{K}$ are vectors of latent Gaussian random variables. For training, we use a single sample ($K = 1$), which is equivalent to variational inference. For testing, we use 5000 samples ($K = 5000$), which is equivalent to importance weighted variational inference, providing a tighter bound on the log-likelihood (Burda et al., 2016). For each $L$-layer VAE, we used a bottom-up variational posterior and top-down generative process with latent variables $\{\mathbf{v}_i\}_{i=1}^{L}$

$$q(\{\mathbf{v}_i\}_{i=1}^{L}|\mathbf{x}) = q(\mathbf{v}_1|\mathbf{x}) \prod_{l=2}^{L} q(\mathbf{v}_l|\mathbf{v}_{l-1}, \mathbf{x}), \quad p(\mathbf{x}, \{\mathbf{v}_i\}_{i=1}^{L}) = p(\mathbf{x}|\{\mathbf{v}_i\}_{i=1}^{L}) \prod_{l=1}^{L-1} p(\mathbf{v}_l|\mathbf{v}_{l+1})$$

with latent spaces of 50 units and each conditional distribution is parameterised by a MLP with two hidden layers of 200 `tanh` units (see Burda et al. (2016); Bauer and Mnih (2021)). We used a factorised Bernoulli likelihood, and factorised Gaussian variational posterior and prior.

We applied the R2-G2 estimator as a single layer within the *decoder* that applies reparameterisation and the linear transformation that follows it (i.e. $\mathbf{W}$ and $\mathbf{V}$ are decoder parameters). We used dynamic binarisation of all three datasets (Salakhutdinov and Murray, 2008). We used the standard train and test splits for MNIST and Fashion-MNIST, and the train and test split from (Burda et al., 2016) for Omniglot. See Table 3 for test lower bounds on the log-likelihood of all datasets after 100,000 steps.

Across all datasets and hierarchical VAEs, we found that the R2-G2 estimator consistently yielded higher test ELBOs than the RT estimator. For three-layer VAEs, we saw substantial gains of 9.05 and 1.17 nats on MNIST and Omniglot respectively (see Figures 2 and 9). These results extend benefits of initial training with Rao-Blackwellised gradients, from Bayesian NNs to Hierarchical VAEs.

## 6 Conclusion

We have presented the R2-G2 estimator as a novel, general-purpose and single-sample gradient estimator for latent Gaussian random variables as the Rao-Blackwellisation of the reparameterisation gradient estimator. Our method is motivated by the widespread usage of Gaussian distributions in probabilistic ML and applications of Rao-Blackwellisation in gradient estimators for latent discrete variables (Liu et al., 2019; Dong et al., 2020; Kool et al., 2020; Paulus et al., 2021).

We theoretically and empirically show that the local reparameterisation trick is an instance of Rao-Blackwellisation and the R2-G2 estimator for linear layers of BNNs with independent weights. It is equivalent to sampling pre-activations in forward computations and updating parameters with a Rao-Blackwellised reparameterisation gradient estimator in backpropagation. This explicates the empirical evidence that the local reparameterisation trick reduces variance of gradients obtained

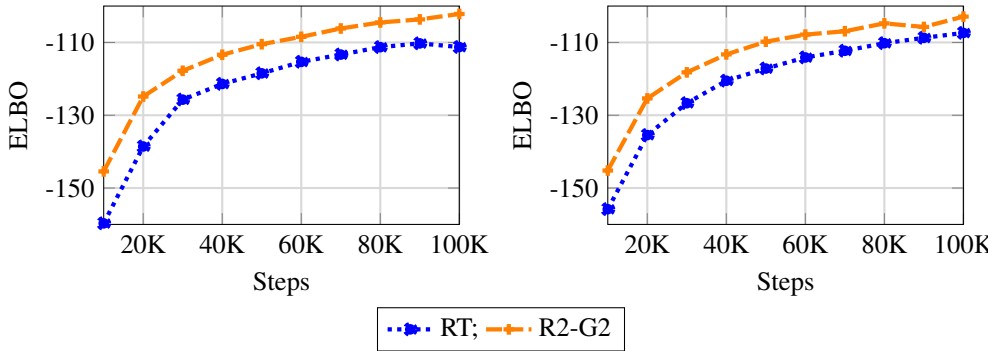

Figure 2: Bounds on log-likelihood v.s. optimisation steps for a three-layer VAE trained on *MNIST* over 5 runs. We compare the bounds on log-likelihoods when training using the reparameterisation (RT) and R2-G2 estimators. Training with the R2-G2 estimator improves bounds on log-likelihood on both the training set (left) and test set (right).

by the global reparameterisation trick as the benefit of Rao-Blackwellisation stated by Theorem 4.3. By casting the local reparameterisation gradient estimator as Rao-Blackwellised gradients for Bayesian MLPs, we showed that initial training with Rao-Blackwellised gradients also yield gains in performance for other models such as Bayesian CNNs and hierarchical VAEs. While performance gains were sometimes modest, they were consistent across models with multiple applications of the reparameterisation trick and particularly prominent in hierarchical VAEs. The main limitation of the R2-G2 estimator is the computational cost from solving Equation 9 and its implicit dependence on how pre-activations are computed. The latter is inherited from model parameterisation. To mitigate the former, we exploited low dimensionality in upper convolutional layers of Bayesian CNNs and latent spaces of VAEs, thereby requiring less iterations of the conjugate gradient algorithm.

While our work has focused on the case where the matrix $\mathbf{V}$ is diagonal, we note that R2-G2 can be extended to the non-diagonal case, such as low-rank covariance matrices (Tomczak et al., 2020), by defining appropriate matrix-vector product functions called within the conjugate gradient algorithm. Aside from likelihoods and ELBOs, we note that the R2-G2 estimator can be applied to other objective functions. A potential extension of our work can be to augment existing gradient estimators for variational inference in the *multi-sample* setting (Roeder et al., 2017; Rainforth et al., 2018; Tucker et al., 2019; Bauer and Mnih, 2021) with the R2-G2 estimator and evaluate its effectiveness. We leave these directions for future work.

# 7 Acknowledgements

We would like to thank Andriy Mnih for helpful discussions and comments on drafts of this paper. Kevin H. Lam gratefully acknowledges his PhD funding from Google DeepMind. Thang D. Bui acknowledges support from the National Computing Infrastructure (NCI) Australia. George Deligiannidis acknowledges support from the Engineering and Physical Sciences Research Council [grant number EP/Y018273/1]. Yee Whye Teh acknowledges support from the Ministry of Digital Development and Information (MDDI) under the Singapore Global AI Visiting Professorship Program (Award No. AIVP-2024-002).

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

# A  Supporting results

This section lists the results supporting the design of the R2-G2 estimator, namely the analytical form of conditional Gaussian distributions and the conjugate gradient algorithm.

**Lemma A.1 ((Eaton, 1983, Proposition 3.13))** *Suppose we have a random vector $\boldsymbol{\epsilon} \sim \mathcal{N}(\mathbf{0}_n, \mathbf{I}_n)$ where $\boldsymbol{\epsilon} \in \mathbb{R}^n$ and a linear map defined as*

$$\mathbf{A} : \mathbb{R}^n \to \mathbb{R}^m, \quad \mathbf{A}(\mathbf{x}) = \mathbf{A}\mathbf{x}$$

*where $\mathbf{A} \in \mathbb{R}^{m \times n}$. Then we have the joint distribution*

$$\begin{bmatrix} \boldsymbol{\epsilon} \\ \mathbf{A}(\boldsymbol{\epsilon}) \end{bmatrix} \sim \mathcal{N}\left( \begin{bmatrix} \mathbf{0}_n \\ \mathbf{0}_m \end{bmatrix}, \begin{bmatrix} \mathbf{I}_n & \mathbf{A}^\top \\ \mathbf{A} & \mathbf{A}\mathbf{A}^\top \end{bmatrix} \right),$$

*and the conditional distribution*

$$\boldsymbol{\epsilon} | \mathbf{A}(\boldsymbol{\epsilon}) = \mathbf{z} \sim \mathcal{N}\left( \mathbf{A}^\top \left(\mathbf{A}\mathbf{A}^\top\right)^\dagger \mathbf{z}, \mathbf{I}_n - \mathbf{A}^\top \left(\mathbf{A}\mathbf{A}^\top\right)^\dagger \mathbf{A} \right)$$

*where $\left(\mathbf{A}\mathbf{A}^\top\right)^\dagger$ is the Moore-Penrose pseudo-inverse of $\mathbf{A}\mathbf{A}^\top$.*

---

**Algorithm 2** Conjugate Gradient Algorithm

---

   **Input:** number of iterations $T$, matrix $\mathbf{A} \in \mathbb{R}^{m \times n}$, column vector $\mathbf{z} \in \mathbb{R}^m$.
   Initialise $\mathbf{x}_0 \in \mathbb{R}^m$.
   Set $\mathbf{r}_0 = \mathbf{A}\mathbf{A}^\top \mathbf{x}_0 - \mathbf{z}$.
   Set $\mathbf{p}_0 = -\mathbf{r}_0$.
   Set $k = 0$.
   **while** $k < T$ and $\mathbf{r}_k \neq \mathbf{0}$ **do**
      Compute $\alpha_k = \frac{\mathbf{r}_k^\top \mathbf{r}_k}{\mathbf{p}_k^\top \mathbf{A}\mathbf{A}^\top \mathbf{p}_k}$.
      Compute $\mathbf{x}_{k+1} = \mathbf{x}_k + \alpha_k \mathbf{p}_k$.
      Compute $\mathbf{r}_{k+1} = \mathbf{r}_k + \alpha_k \mathbf{A}\mathbf{A}^\top \mathbf{p}_k$.
      Compute $\beta_{k+1} = \frac{\mathbf{r}_{k+1}^\top \mathbf{r}_{k+1}}{\mathbf{r}_k^\top \mathbf{r}_k}$.
      Compute $\mathbf{p}_{k+1} = -\mathbf{r}_{k+1} + \beta_{k+1} \mathbf{p}_k$.
      Increment $k$ by 1.
   **end while**
   **Output:** $\mathbf{x}_T$.

---

# B  Proof for Proposition 4.2

We first prove that the R2-G2 estimator is an unbiased estimator of $\nabla_{\boldsymbol{\theta}} \mathbb{E}_{q_{\boldsymbol{\theta}}}[\ell]$. Recall that

$$\widehat{\nabla_{\boldsymbol{\theta}}\ell}^{R2\text{-}G2} = \mathbb{E}_{\tilde{q}_0}\left[\widehat{\nabla_{\boldsymbol{\theta}}\ell}^{RT}\right].$$

We note that $\widehat{\nabla_{\boldsymbol{\theta}}\ell}^{R2\text{-}G2}$ is a conditional expectation with $\mathbf{W} \cdot g = \mathbf{z}$ as the conditioning variable. Then we have

$$
\begin{aligned}
\mathbb{E}_{q_{\mathbf{z}}}\left[\widehat{\nabla_{\boldsymbol{\theta}}\ell}^{R2\text{-}G2}\right] &= \mathbb{E}_{q_{\mathbf{z}}}\left[\mathbb{E}_{\tilde{q}_0}\left[\widehat{\nabla_{\boldsymbol{\theta}}\ell}^{RT}\right]\right] \\
&= \mathbb{E}_{q_0}\left[\widehat{\nabla_{\boldsymbol{\theta}}\ell}^{RT}\right] \\
&= \nabla_{\boldsymbol{\theta}}\mathbb{E}_{q_{\boldsymbol{\theta}}}[\ell]
\end{aligned}
$$

where the second equality follows from the law of iterated expectations. Since the reparameterisation gradient estimator $\widehat{\nabla_{\boldsymbol{\theta}}\ell}^{RT}$ is an unbiased estimator of $\nabla_{\boldsymbol{\theta}}\mathbb{E}_{q_{\boldsymbol{\theta}}}[\ell]$, it follows that the $R2\text{-}G2$ estimator $\widehat{\nabla_{\boldsymbol{\theta}}\ell}^{R2\text{-}G2}$ is also an unbiased estimator of $\nabla_{\boldsymbol{\theta}}\mathbb{E}_{q_{\boldsymbol{\theta}}}[\ell]$.

We are left to show that the R2-G2 estimator has less variance than the reparameterisation gradient estimator. We can write

$$
\begin{aligned}
\mathbb{E}_{q_{\mathbf{z}}}\left[\left\|\widehat{\nabla_{\boldsymbol{\theta}}\ell}^{R2\text{-}G2} - \nabla_{\boldsymbol{\theta}}\mathbb{E}_{q_{\boldsymbol{\theta}}}[\ell]\right\|^2\right] &= \mathbb{E}_{q_{\mathbf{z}}}\left[\left\|\mathbb{E}_{\tilde{q}_0}\left[\widehat{\nabla_{\boldsymbol{\theta}}\ell}^{RT}\right] - \nabla_{\boldsymbol{\theta}}\mathbb{E}_{q_{\boldsymbol{\theta}}}[\ell]\right\|^2\right] \\
&= \mathbb{E}_{q_{\mathbf{z}}}\left[\left\|\mathbb{E}_{\tilde{q}_0}\left[\widehat{\nabla_{\boldsymbol{\theta}}\ell}^{RT} - \nabla_{\boldsymbol{\theta}}\mathbb{E}_{q_{\boldsymbol{\theta}}}[\ell]\right]\right\|^2\right] \\
&\leq \mathbb{E}_{q_{\mathbf{z}}}\left[\mathbb{E}_{\tilde{q}_0}\left[\left\|\widehat{\nabla_{\boldsymbol{\theta}}\ell}^{RT} - \nabla_{\boldsymbol{\theta}}\mathbb{E}_{q_{\boldsymbol{\theta}}}[\ell]\right\|^2\right]\right] \\
&= \mathbb{E}_{q_0}\left[\left\|\widehat{\nabla_{\boldsymbol{\theta}}\ell}^{RT} - \nabla_{\boldsymbol{\theta}}\mathbb{E}_{q_{\boldsymbol{\theta}}}[\ell]\right\|^2\right]
\end{aligned}
$$

where the inequality results from using Jensen's inequality, and the last equality comes from the law of iterated expectations.

# C  Proof of Theorem 4.3

Suppose we have a BNN linear layer where weights are independent Gaussian random variables. Given an input $\mathbf{x} \in \mathbb{R}^n$, the pre-activations and parameters of these linear layers are respectively given by

$$
\mathbf{z} = \begin{bmatrix} \mathbf{x}^\top g(\boldsymbol{\epsilon}^{(1)}, \boldsymbol{\theta}^{(1)}) \\ \vdots \\ \mathbf{x}^\top g(\boldsymbol{\epsilon}^{(m)}, \boldsymbol{\theta}^{(m)}) \end{bmatrix} \in \mathbb{R}^m
$$

where $\boldsymbol{\theta}^{(i)} = \{\boldsymbol{\mu}^{(i)}, \boldsymbol{\tau}^{(i)}\}$ for $i = 1, \ldots, m$. We proceed by considering a fixed $i$.

By setting $\mathbf{V}^{(i)} = (\boldsymbol{\Sigma}^{(i)})^{\frac{1}{2}} = \left(\texttt{diag}(\boldsymbol{\tau}^{(i)})\right)^{\frac{1}{2}}$ and $\mathbf{W} = \mathbf{x}^\top$ within Equation 6, the reparameterisation gradient estimator is given by

$$\widehat{\nabla_{\boldsymbol{\theta}^{(i)}}[\ell]}^{RT} = \left(\frac{\partial \tilde{\ell}}{\partial z_i} \cdot \mathbf{x}^\top \cdot \left[\; \mathbf{I}_n \;\middle|\; \tfrac{1}{2}(\boldsymbol{\Sigma}^{(i)})^{-\frac{1}{2}} \odot (\mathbf{1}_n(\boldsymbol{\epsilon}^{(i)})^\top) \;\right]\right)^\top.$$

We proceed with the idea of Rao-Blackwellisation from the R2-G2 estimator by conditioning on the pre-activation $z_i = \mathbf{x}^\top \cdot g(\boldsymbol{\epsilon}^{(i)}, \boldsymbol{\theta}^{(i)}) \in \mathbf{z}$. This amounts to setting $\mathbf{A} = \mathbf{x}^\top \mathbf{V}^{(i)}$ in Definition 4.1.

We note that $\mathbf{x}^\top \cdot g(\boldsymbol{\epsilon}^{(i)}, \boldsymbol{\theta}^{(i)}) = z_i$ is equivalent to $\mathbf{A}\boldsymbol{\epsilon}^{(i)} = z_i - \mathbf{x}^\top \boldsymbol{\mu}^{(i)} = \tilde{z}_i$. Denote $\tilde{\boldsymbol{\epsilon}}^{(i)}$ as $\boldsymbol{\epsilon}^{(i)}|\mathbf{A}\boldsymbol{\epsilon}^{(i)} = \tilde{z}_i$ with distribution denoted $\tilde{q}_0^{(i)}$. Using Lemma A.1, we can write the distribution

$\tilde{\boldsymbol{\epsilon}}^{(i)} \sim \tilde{q}_0^{(i)}$ as

$$\tilde{q}_0^{(i)} = \mathcal{N}\left((\boldsymbol{\Sigma}^{(i)})^{\frac{1}{2}}\mathbf{x}\left(\frac{\tilde{z}_i}{\sum_{j=1}^n x_j^2\left(\sigma_j^{(i)}\right)^2}\right), \mathbf{I}_n - (\boldsymbol{\Sigma}^{(i)})^{\frac{1}{2}}\mathbf{x}\left(\sum_{j=1}^n x_j^2\left(\sigma_j^{(i)}\right)^2\right)^{-1}\mathbf{x}^\top(\boldsymbol{\Sigma}^{(i)})^{\frac{1}{2}}\right).$$

Define the transformation of a vector to a diagonal matrix as

$$D : \mathbb{R}^n \to \mathbb{R}^{n\times n}, \quad D(\mathbf{x}) = \sum_{k=1}^n e_k^\top \mathbf{x} e_k e_k^\top$$

where $e_k \in \mathbb{R}^n$ is the $k$-th standard basis vector. Using the linearity of expectations, we have

$$\frac{1}{2}\mathbf{x}^\top \cdot \mathbb{E}_{\tilde{q}_0^{(i)}}\left[(\boldsymbol{\Sigma}^{(i)})^{-\frac{1}{2}} \odot \mathbf{1}_n(\boldsymbol{\epsilon}^{(i)})^\top\right] = \frac{1}{2}\mathbf{x}^\top \cdot \left[(\boldsymbol{\Sigma}^{(i)})^{-\frac{1}{2}} \odot \mathbf{1}_n\left(\mathbb{E}_{\tilde{q}_0^{(i)}}[\boldsymbol{\epsilon}^{(i)}]\right)^\top\right]$$

$$= \frac{1}{2}\left(\frac{\tilde{z}_i}{\sum_{j=1}^n x_j^2\left(\sigma_j^{(i)}\right)^2}\right)\mathbf{x}^\top \cdot \left[(\boldsymbol{\Sigma}^{(i)})^{-\frac{1}{2}} \odot (\mathbf{1}_n\mathbf{x}^\top(\boldsymbol{\Sigma}^{(i)})^{\frac{1}{2}})\right]$$

$$= \frac{1}{2}\left(\frac{\tilde{z}_i}{\sum_{j=1}^n x_j^2\left(\sigma_j^{(i)}\right)^2}\right)\mathbf{x}^\top D\left(\mathbf{x}\right)$$

$$= \frac{1}{2}\left(\frac{z_i - \sum_{j=1}^n x_j\mu_j^{(i)}}{\sum_{j=1}^n x_j^2\left(\sigma_j^{(i)}\right)^2}\right)(\mathbf{x} \odot \mathbf{x})^\top$$

$$\stackrel{d}{=} \frac{1}{2}\left(\sum_{j=1}^n x_j^2\left(\sigma_j^{(i)}\right)^2\right)^{-\frac{1}{2}}\xi_i\left(\mathbf{x} \odot \mathbf{x}\right)^\top$$

where $\mathbf{x}^\top D\left(\mathbf{x}\right) = (\mathbf{x} \odot \mathbf{x})^\top$ is a row vector with entries $\left\{x_j^2\right\}_{j=1}^n$ and applying the reparameterisation trick gives us

$$z_i \stackrel{d}{=} \sum_{j=1}^n x_j\mu_j^{(i)} + \left(\sum_{i=1}^n x_j^2\left(\sigma_j^{(i)}\right)^2\right)^{\frac{1}{2}}\xi_i \sim \mathcal{N}\left(\sum_{j=1}^n x_j\mu_j^{(i)}, \sum_{j=1}^n x_j^2\left(\sigma_j^{(i)}\right)^2\right)$$

for $\xi_i \sim \mathcal{N}(0, 1)$. Since we also have $\mathbf{x}^\top \cdot \mathbb{E}_{\tilde{q}_0^{(i)}}[\mathbf{I}_n] = \mathbf{x}^\top$, the Jacobian of $z_i = \mathbf{x}^\top \cdot g(\boldsymbol{\epsilon}^{(i)}, \boldsymbol{\theta}^{(i)})$ is

$$J_{z_i}(\boldsymbol{\theta}^{(i)}) = \mathbf{x}^\top \cdot \mathbb{E}_{\tilde{q}_0^{(i)}}[J_{g^{(i)}}]$$

$$\stackrel{d}{=} \left[\mathbf{x}^\top \mid \frac{1}{2}\left(\sum_{j=1}^n x_j^2\left(\sigma_j^{(i)}\right)^2\right)^{-\frac{1}{2}}\xi_i\left(\mathbf{x} \odot \mathbf{x}\right)^\top\right].$$

Hence, we have

$$\widehat{\nabla_{\boldsymbol{\theta}^{(i)}}\ell}^{R2\text{-}G2} \stackrel{d}{=} \widehat{\nabla_{\boldsymbol{\theta}^{(i)}}\ell}^{LRT}.$$

This shows that the local reparameterisation gradient estimator is equivalent in distribution to a Rao-Blackwellised reparameterisation gradient estimator.

We are left to show that the local reparameterisation estimator has lower variance than the global reparameterisation estimator. We have

$$
\mathbb{E}_{\tilde{q}_{z_i}}\left[\left\|\widehat{\nabla_{\boldsymbol{\theta}^{(i)}}\ell}^{LRT} - \nabla_{\boldsymbol{\theta}^{(i)}}\mathbb{E}_{q_{\boldsymbol{\theta}^{(i)}}}[\ell]\right\|^2\right] = \mathbb{E}_{\tilde{q}_{z_i}}\left[\left\|\widehat{\nabla_{\boldsymbol{\theta}^{(i)}}\ell}^{R2\text{-}G2} - \nabla_{\boldsymbol{\theta}^{(i)}}\mathbb{E}_{q_{\boldsymbol{\theta}^{(i)}}}[\ell]\right\|^2\right]
$$

$$
= \mathbb{E}_{\tilde{q}_{z_i}}\left[\left\|\mathbb{E}_{\tilde{q}_0}\left[\widehat{\nabla_{\boldsymbol{\theta}^{(i)}}\ell}^{RT}\right] - \nabla_{\boldsymbol{\theta}^{(i)}}\mathbb{E}_{q_{\boldsymbol{\theta}^{(i)}}}[\ell]\right\|^2\right]
$$

$$
= \mathbb{E}_{\tilde{q}_{z_i}}\left[\left\|\mathbb{E}_{\tilde{q}_0}\left[\widehat{\nabla_{\boldsymbol{\theta}^{(i)}}\ell}^{RT} - \nabla_{\boldsymbol{\theta}^{(i)}}\mathbb{E}_{q_{\boldsymbol{\theta}^{(i)}}}[\ell]\right]\right\|^2\right]
$$

$$
\leq \mathbb{E}_{\tilde{q}_{z_i}}\left[\mathbb{E}_{\tilde{q}_0}\left[\left\|\widehat{\nabla_{\boldsymbol{\theta}^{(i)}}\ell}^{RT} - \nabla_{\boldsymbol{\theta}^{(i)}}\mathbb{E}_{q_{\boldsymbol{\theta}^{(i)}}}[\ell]\right\|^2\right]\right]
$$

$$
= \mathbb{E}_{q_0^{(i)}}\left[\left\|\widehat{\nabla_{\boldsymbol{\theta}^{(i)}}\ell}^{RT} - \nabla_{\boldsymbol{\theta}^{(i)}}\mathbb{E}_{q_{\boldsymbol{\theta}^{(i)}}}[\ell]\right\|^2\right]
$$

where the inequality results from using Jensen's inequality and the last equality comes from the law of iterated expectations.

## D  Computation of Conditional Mean

Recall that the image and kernel of a linear map $\mathbf{T} : \mathbb{R}^n \to \mathbb{R}^m$ are defined respectively as $\mathtt{im}(\mathbf{T}) = \{\mathbf{Tv} : \mathbf{v} \in \mathbb{R}^n\}$ and $\mathtt{ker}(\mathbf{T}) = \{\mathbf{v} : \mathbf{Tv} = \mathbf{0}\}$. Since $\mathbf{A}\boldsymbol{\epsilon} \in \mathtt{im}(\mathbf{A}) = \mathtt{im}(\mathbf{A}\mathbf{A}^\top)$, we note that Equation 9 has solutions in the form

$$
\left(\mathbf{A}\mathbf{A}^\top\right)^\dagger \mathbf{A}\boldsymbol{\epsilon} + (\mathbf{I}_m - \left(\mathbf{A}\mathbf{A}^\top\right)^\dagger \mathbf{A}\mathbf{A}^\top)\mathbf{y}
$$

for any $\mathbf{y} \in \mathbb{R}^m$ (see James (1978); Planitz (1979)). Here, $\left(\mathbf{A}\mathbf{A}^\top\right)^\dagger \mathbf{A}\boldsymbol{\epsilon} \in \mathtt{im}(\mathbf{A}\mathbf{A}^\top) = \mathtt{im}(\mathbf{A})$ and $(\mathbf{I}_m - \left(\mathbf{A}\mathbf{A}^\top\right)^\dagger \mathbf{A}\mathbf{A}^\top)\mathbf{y} \in \mathtt{ker}(\mathbf{A}\mathbf{A}^\top) = \mathtt{ker}(\mathbf{A}^\top)$. That is, we can recast the matrix-vector product $\left(\mathbf{A}\mathbf{A}^\top\right)^\dagger \mathbf{A}\boldsymbol{\epsilon}$ as a solution to Equation 9, up to an additive term from $\mathtt{ker}(\mathbf{A}^\top)$. For any solution $\boldsymbol{\beta}^*$ to Equation 9, it then follows that

$$
\mathbf{A}^\top \boldsymbol{\beta}^* = \mathbf{A}^\top \left(\mathbf{A}\mathbf{A}^\top\right)^\dagger \mathbf{A}\boldsymbol{\epsilon} = \boldsymbol{\epsilon}^*.
$$

Table 4: Optimisation hyperparameters and running times for experiments.

| Model | Estimator | Steps | Learning Rate | Steps/s | Run Time Limit (hours) |
|-------|-----------|-------|---------------|---------|------------------------|
| Bayesian MLP | R2-G2 | 15,000 | 0.0001 | 13.57 | 0.5 |
| Bayesian MLP | LRT | 15,000 | 0.0001 | 48.25 | 0.5 |
| Bayesian MLP | RT | 15,000 | 0.0001 | 10.52 | 0.5 |
| Bayesian CNN | R2-G2 | 12,500 | 0.0001 | 0.68 | 10 |
| Bayesian CNN | RT | 12,500 | 0.0001 | 2.31 | 5 |
| One-layer VAE | R2-G2 | 100,000 | 0.0003 | 47.96 | 1 |
| One-layer VAE | RT | 100,000 | 0.0003 | 127.66 | 1 |
| Two-layer VAE | R2-G2 | 100,000 | 0.0003 | 20.22 | 1.5 |
| Two-layer VAE | RT | 100,000 | 0.0003 | 78.72 | 1.5 |
| Three-layer VAE | R2-G2 | 100,000 | 0.0003 | 17.11 | 2.5 |
| Three-layer VAE | RT | 100,000 | 0.0003 | 48.41 | 2.5 |

# E  Experiment details

**Compute resources**    All experiments were run on a single NVIDIA V100 GPU.

**Architectures**    Model architectures for BNN and VAE experiments can be found in Section 5. We excluded batch normalisation and dropout layers from model architectures.

**Optimisation**    We used a batch size of 80 and the Adam optimiser for all experiments Kingma and Ba (2015). We do not add regularisation such as weight decay, dropout or batch normalisation layers. Other optimisation parameters are listed in Table 4.

For consistency with work on gradient estimators, we report the number of optimisation/SGD steps. For BNN experiments on the MNIST and CIFAR-10 datasets, this is equivalent to training with a batch size of 80 for 20 epochs.

# F  Comparison of Computational Cost of Gradient Estimators

In general, the worst-case complexity of computational costs from the conjugate gradient (CG) algorithm is $\mathcal{O}(m^3)$ when one is attempting to invert a $m \times m$ dense matrix $\mathbf{A}$ without knowing its structure. Here the worst-case would be running $m$ iterations of CG with each iteration requiring $m^2$ flops for a matrix-vector product.

In the mean-field setting, which is the main focus of our work, we know additional structure about the matrix $\mathbf{A}$. Specifically, it can factorise as $\mathbf{A} = \mathbf{W}\mathbf{V}\mathbf{V}^{\top}\mathbf{W}^{\top}$ where $\mathbf{V} \in \mathbb{R}^{n \times n}$ is diagonal and $\mathbf{W} \in \mathbb{R}^{m \times n}$. Moreover, we know the rank of $\mathbf{V}$ and the maximum rank of $\mathbf{W}$, so CG only needs to be run for $k = \min(m, n)$ iterations at most, by using the property of ranks that $\text{rank}(AB) \leq min(\text{rank}(A), \text{rank}(B))$ for matrices $A, B$.

A forward pass for a linear layer using the global reparameterisation trick would use a total of $(2m + 1)n$ flops for matrix-vector products. Using the R2-G2 estimator runs at most $k$ iterations of the CG algorithm with each iteration requiring $(2m + 1)n$ flops for matrix-vector products, so an additional $(2m + 1)nk$ flops would be incurred at most.

For BNNs, we would have $n > m^2$ since there are more weights than pre-activations, so the cost complexity would be $\mathcal{O}(n)$ for a forward pass using the global reparameterisation trick and an additional $\mathcal{O}(m^2n)$ for R2-G2. For VAEs, we would have $m > n$ since we map low-dimensional latents to a higher-dimensional space, so the cost complexity would be $\mathcal{O}(m)$ for a forward pass using the global reparameterisation trick and an additional $\mathcal{O}(mn^2)$ for R2-G2.

# G  Additional plots of gradient variance

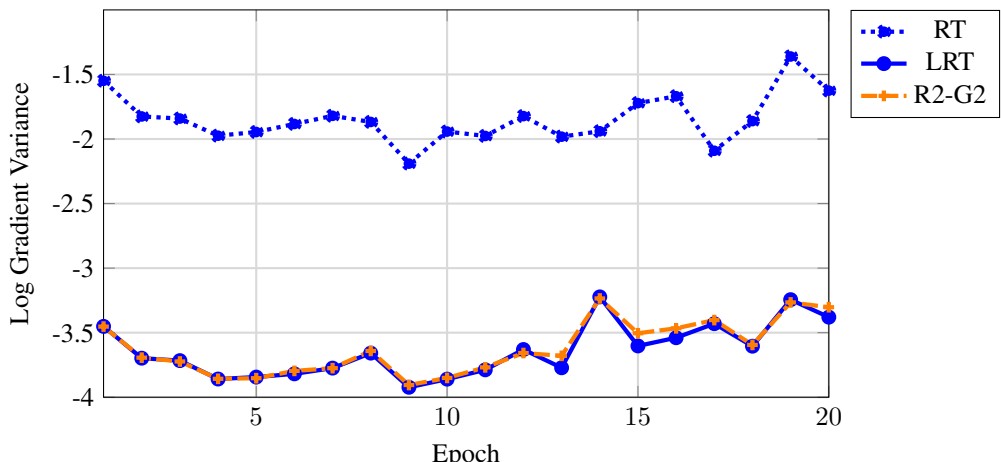

Figure 3: Log gradient variance v.s. epoch for the bottom layer of a Bayesian MLP trained on *MNIST* over 5 runs. We compare the variance of gradients when training using the reparameterisation (RT), local reparameterisation (LRT) and R2-G2 estimators.

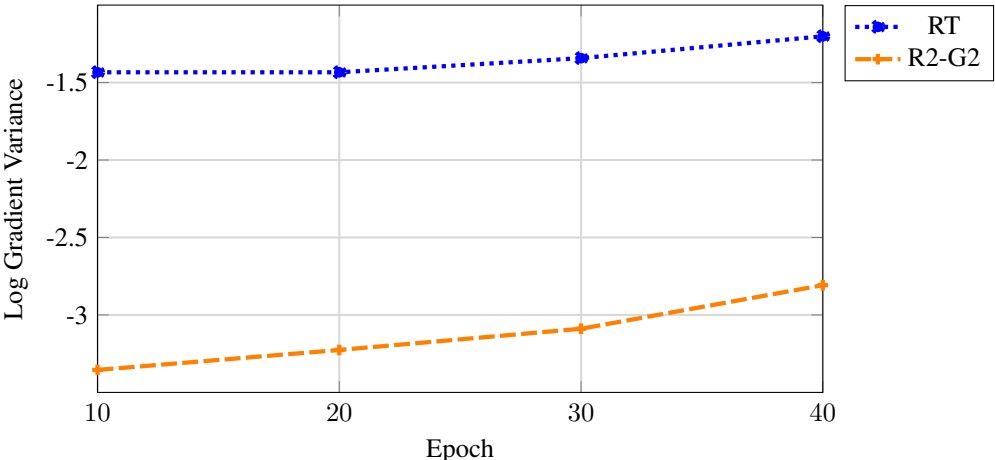

Figure 4: Log gradient variance v.s. epoch for the 8-th convolutional layer of a Bayesian CNN trained on *CIFAR-10* over 5 runs. We compare the variance of gradients when training using the reparameterisation (RT) and R2-G2 estimators.

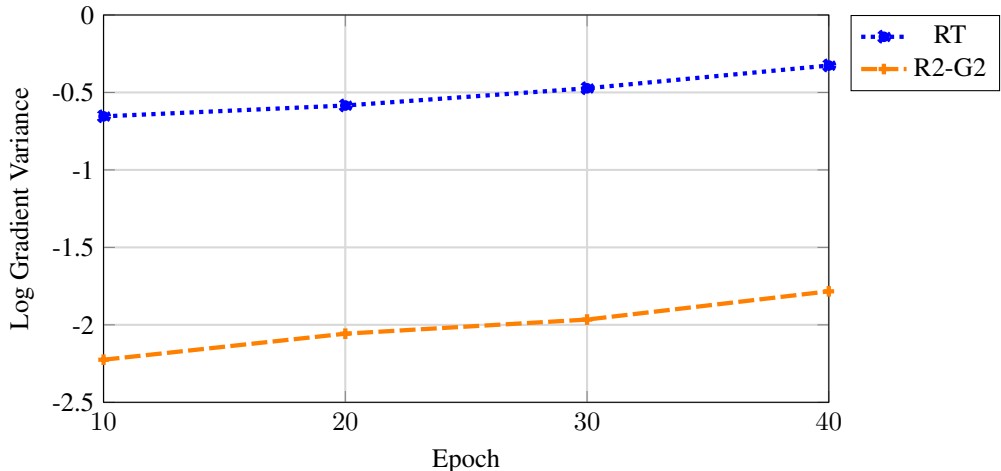

Figure 5: Log gradient variance v.s. epoch for the 5-th convolutional layer of a Bayesian CNN trained on *CIFAR-10* over 5 runs. We compare the variance of gradients when training using the reparameterisation (RT) and R2-G2 estimators.

## H  Experiments on single-layer Variational Autoencoders

Table 5: Test variational lower bounds for VAEs using the R2-G2 and Reparameterisation (RT) estimators. Higher is better. Error bars denote $\pm 1.96$ standard errors $(\sigma/\sqrt{5})$ over 5 runs.

| # VAE Layers | Estimator | MNIST | Omniglot | Fashion-MNIST |
|---|---|---|---|---|
| 1 | R2-G2 | $-94.39 \pm 0.42$ | $\mathbf{-117.61 \pm 2.12}$ | $-238.65 \pm 0.26$ |
|   | RT | $\mathbf{-94.22 \pm 0.24}$ | $-117.64 \pm 2.12$ | $\mathbf{-238.64 \pm 0.25}$ |

The R2-G2 estimator yielded limited gains on performance for single-layer VAEs. In this setting, the R2-G2 estimator only impacts the optimisation of the encoder. This motivates our experiments for hierarchical VAEs where we only apply the R2-G2 estimator within the decoder (i.e. $\mathbf{W}$ and $\mathbf{V}$ are both matrix parameters in the decoder).

## I  Additional plots of bounds on log-likelihood

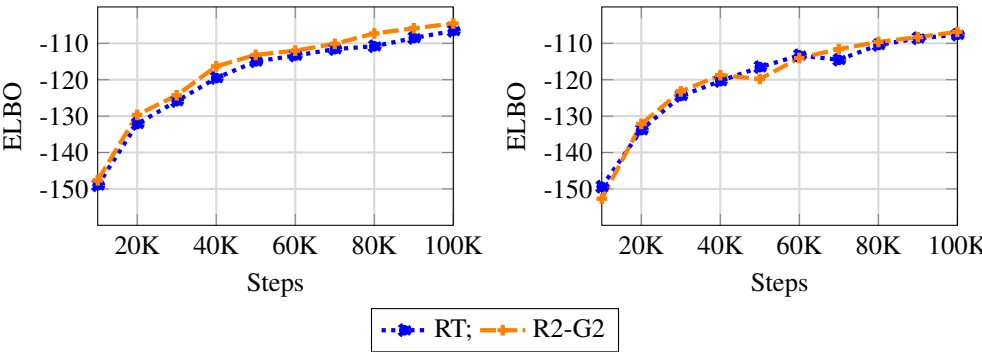

Figure 6: Bounds on log-likelihood v.s. optimisation steps for a two-layer VAE trained on *MNIST* over 5 runs. We compare the bounds on log-likelihoods when training using the reparameterisation (RT) and R2-G2 estimators on both the training set (left) and test set (right).

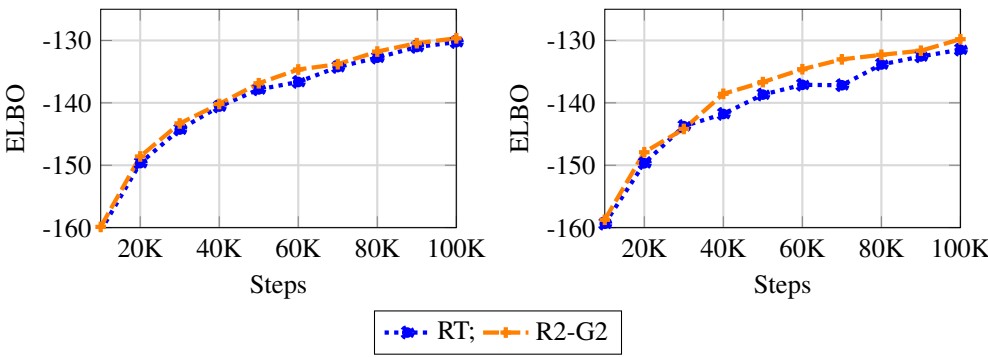

Figure 7: Bounds on log-likelihood v.s. optimisation steps for a two-layer VAE trained on *Omniglot* over 5 runs. We compare the bounds on log-likelihoods when training using the reparameterisation (RT) and R2-G2 estimators on both the training set (left) and test set (right).

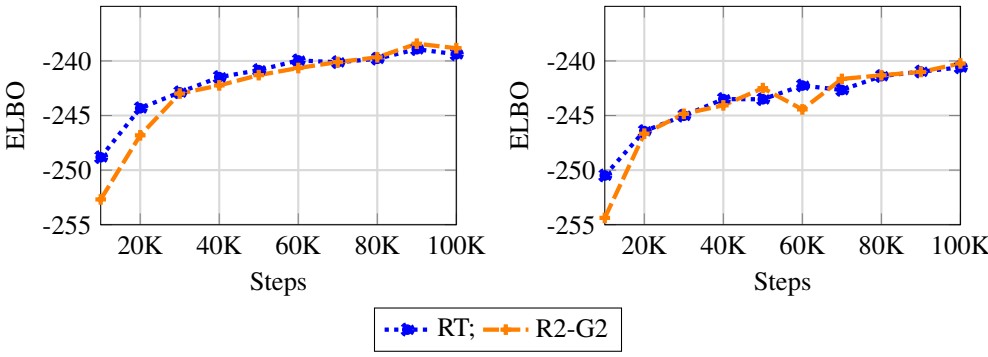

Figure 8: Bounds on log-likelihood v.s. optimisation steps for a two-layer VAE trained on *Fashion-MNIST* over 5 runs. We compare the bounds on log-likelihoods when training using the reparameterisation (RT) and R2-G2 estimators on both the training set (left) and test set (right).

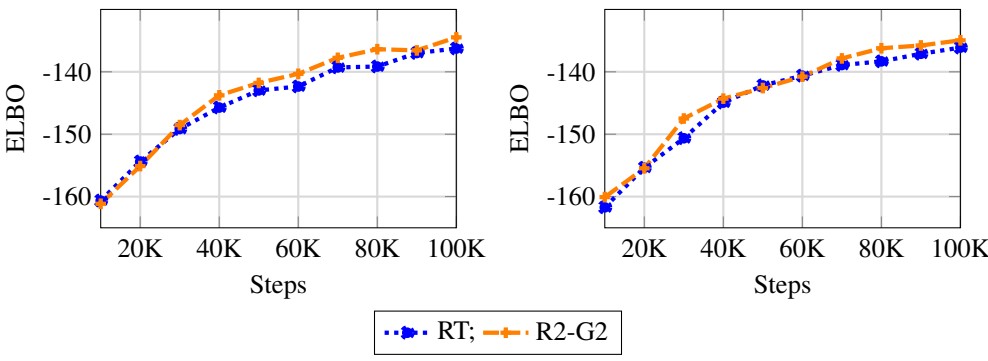

Figure 9: Bounds on log-likelihood v.s. optimisation steps for a three-layer VAE trained on *Omniglot* over 5 runs. We compare the bounds on log-likelihoods when training using the reparameterisation (RT) and R2-G2 estimators on both the training set (left) and test set (right).

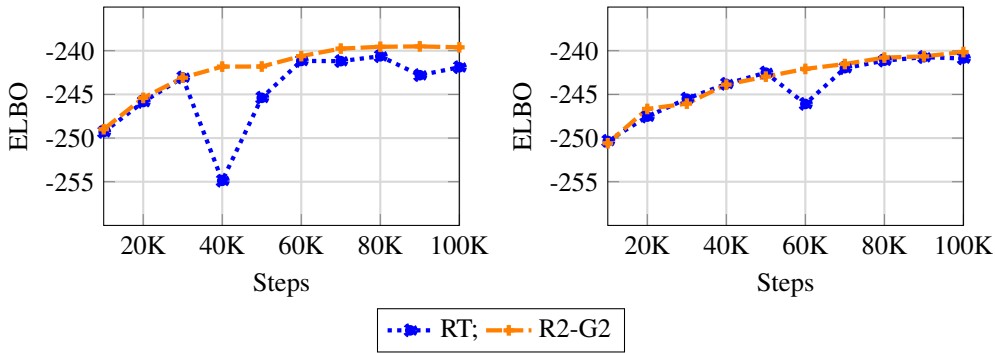

Figure 10: Bounds on log-likelihood v.s. optimisation steps for a three-layer VAE trained on *Fashion-MNIST* over 5 runs. We compare the bounds on log-likelihoods when training using the reparameterisation (RT) and R2-G2 estimators on both the training set (left) and test set (right).

## J  Licenses

Codebases:

- Convex Potential Flows: Universal Probability Distributions with Optimal Transport and Convex Optimization (Huang et al., 2021): MIT license.

Datasets:

- MNIST (LeCun et al., 2010): Creative Commons Attribution-Share Alike 3.0 license
- CIFAR-10 (Krizhevsky and Hinton, 2009): MIT license
- Fashion-MNIST (Xiao et al., 2017): MIT license
- Omniglot (Lake et al., 2015): MIT license

The conjugate gradient algorithm for each BNN and VAE architecture is modified from the above codebase. All experiments are performed on the above datasets.

