# OpenReview forum: "Rao-Blackwellised Reparameterisation Gradients"
_NeurIPS.cc/2025/Conference — NeurIPS 2025 poster_

### Official Review · Reviewer_ZPWE · 2025-06-25

**Clarity:** 2
**Significance:** 3
**Originality:** 3
**Rating:** 5
**Confidence:** 4

**Summary:**

The authors discuss the problem of estimating stochastic gradients of a function, specifically `\nabla_\theta \mathbb{E}_{q_\theta}[f(\theta, \dots)]`, which are omnipresent in many applications of approximate inference for neural networks. For this, the authors revise the reparameterisation trick and present a Rao-Blackwellised reparameterisation applicable if $q_\theta$ (e.g., the variational family) follows a mean-field Gaussian distribution. This new reparameterisation is then shown to be equivalent to *local* reparameterisation under certain models and empirically (as well as theoretically) reduces variance in the gradient estimates compared to standard reparameterisation. The authors also show some slight improvements in performance on some basic image classification tasks (MNIST and CIFAR-10).

**Questions:**

- Why make the assumption that $v$ is mean-field Gaussian, if you could also derive the distribution for correlated $v$?
- What is the effect of making all of those independence assumptions?
- What is $\text{im}(X)$ and $\ker(X)$?
- What is the computational cost in FLOPS?
- How does this approach scale and perform in more interesting settings (e.g., ImageNet)?
- Why doesn't the convergence improve?

**Ethical Concerns:**

["NO or VERY MINOR ethics concerns only"]

**Final Justification:**

The authors have sufficiently addressed my questions and concerns.

**Limitations:**

The authors discuss some of the limitations of the work in the conclusion section.

**Paper Formatting Concerns:**

There are no formatting concerns.

**Quality:**

2

**Strengths And Weaknesses:**

### Strengths
The idea is novel and methodologically sound. The work involves various rather technical steps, and I appreciate the effort in trying to make the approach scalable. The approach clearly enjoys variance reduction and is convincing in this regard.

### Weaknesses
Unfortunately, the paper is not very well written. Many parts feel unclear at first, and the reader needs to fill in the gaps to understand what the authors are doing or trying to convey. This makes the reading rather unpleasant.

Secondly, the performance improvements are marginal at best. I don't think this is a problem, as I would not necessarily expect big performance gains, but I was surprised to see that the convergence is not improved even though the variance is substantially reduced. This makes me wonder to what extent the high additional complexity is justified, as little gains can effectively be observed.

Lastly, the evaluation is only on toy data by today's standards. I would have expected to see an evaluation in more interesting contexts.

### Additional Comments

Below is a list of additional comments that are not necessarily questions.

- Line 14, the authors say that "latent variables [...] enable us to embed inductive biases like uncertainty in the model specification". I do not understand what this is meant to say. I agree that LVs can help encode prior assumptions, but those aren't necessarily linked to the model's inductive biases. I also do not understand how uncertainty in the model specification is an inductive bias. I see that the inductive bias will affect the distribution of uncertainty over the model specification, but this paragraph seems somewhat convoluted.
- Line 16, the authors say LV appears in "modelling tasks, including Bayesian inference". I do not see how Bayesian inference is a modelling task.
- I generally found the distinction between estimators and "single-sample" estimators confusing. While the reparameterization trick is often used with a single Monte-Carlo sample, I do not see that there is a need to distinguish it from other estimators, as this is a hyperparameter. It is very much conceivable to use more than one sample.
- Eq. 3 inherently assumes that $v \sim g(\epsilon^{(i)}, \theta^{(i)})$ with $v \in \mathbb{R}^n$ is distributed by a mean-field Gaussian. This is a rather strong assumption that is not made explicit. As for the current version, this means the authors assume independence in $v$ and in $z$, which seems very strong.
- Line 102, $\tau$ is not introduced, and it is unclear how this relates to Eq. 3. Additional confusion happens later on where $\tau$ is used.
- Line 121, typo "is" -> "are"
- Paragraph 118 - 124, I believe it would be worth mentioning that the "global" reparameterisation trick is in fact not limited to the Gaussian family but can be applied through inverse CDF sampling to any distribution.
- Section 4.2., I had a really hard time piecing things together in this section. For example, $\text{im}(X)$ and $\ker(X)$ are never introduced. The solution of Eq.9 trivially results from Planitz, but it is hard to understand how this now relates to the goal we are after here. Moreover, it is unclear if $\epsilon^\star$ refers to the mean of the distribution over $\epsilon$ as in Sec. 4.1 or not, as it is referred to as the "fitted noise" and $\beta^\star$ is not the mean of the distribution of $\beta$ but the solution to the quadratic program.
- I believe a more detailed cost estimate of the R2-G2 estimator is necessary to fully characterise the computational cost (ideally in O notation and FLOPS) and compare to vanilla reparameterisation.

---

> ### Author Rebuttal · Authors · 2025-07-30
>
> We thank the reviewer for their review and appreciate their constructive feedback on our work. Please see our responses to each point raised individually below.
>
> ### Weaknesses
> > Many parts feel unclear at first.
>
> Please see our proposed changes in the responses below. We hope these changes improve the reading of our work and make it more accessible to readers.
>
> ### Comments
> > Line 14, the authors say that "latent variables [...] enable us to embed inductive biases like uncertainty in the model specification". I do not understand what this is meant to say.
> > Line 16, the authors say LV appears in "modelling tasks, including Bayesian inference". I do not see how Bayesian inference is a modelling task.
>
> [A] We agree that the language can be expressed better here. In the next version, we propose to change line 14 to "latent variables [...] enable us to embed prior assumptions into models, such as uncertainty in model parameters [...]" and line 16 to “modelling tasks, including variational inference  [...]”.
>
> > I generally found the distinction between estimators and "single-sample" estimators confusing. While the reparameterization trick is often used with a single Monte-Carlo sample, I do not see that there is a need to distinguish it from other estimators, as this is a hyperparameter. It is very much conceivable to use more than one sample.
>
> [B] We note the distinction between single-sample estimators and more general (potentially multi-sample) estimators is crucial on two fronts.
>
> First, the distinction helps distinguish work on single-sample gradient estimators, such as [1] and our work, from the wide literature on gradient estimators that use multiple samples and control variates to compute the expected gradient.
>
> Second, it clarifies where Rao-Blackwellisation is actually applied. One application of R2-G2 will only reduce the gradient variance from one set of Gaussian variables. While applicable to each sample in a mini-batch, it remains a sample-level procedure that is parallelised. On the other hand, multi-sample estimators focus on the expected gradient, so Rao-Blackwellising this implies conditioning over a sample of $K>1$ pre-activation vectors per sample, which is not what we do.
>
> > Line 102, $\tau^{(i)}$ is not introduced, and it is unclear how this relates to Eq. 3. Additional confusion happens later on where $\tau^{(i)}$ is used.
>
> [C] $\tau^{(i)}$ defines a vector of variance parameters in the same way as $\tau$ introduced in Line 86. For clarity, we propose to explicitly state $\tau^{(i)} = ((\sigma_{1}^{(i)})^2, \dots, (\sigma_{n}^{(i)})^2)$ within the local reparameterisation trick paragraph in Section 3.
>
> > Paragraph 118 - 124, I believe it would be worth mentioning that the "global" reparameterisation trick is in fact not limited to the Gaussian family but can be applied through inverse CDF sampling to any distribution.
>
> Great suggestion! We propose to add this in the next version.
>
> ### Questions
> > Why make the assumption that is mean-field Gaussian, if you could also derive the distribution for correlated?
> > What is the effect of making all of those independence assumptions?
>
> [D] Great question! We have focused on the independent case in this work for two reasons.
> First, we wanted to establish a first connection between the classical idea of Rao-Blackwellisation from statistics and the local reparameterisation trick (which enjoys low-variance gradients in practice but can only be derived for mean-field BNN linear layers). Second, we wanted to assess the benefits and limitations from training with Rao-Blackwellised gradients would benefit probabilistic models in this simple setting before investigating the more difficult setting of correlated Gaussians in future work. This uncovered sizeable gains for hierarchical VAEs and limitations for single-layer VAEs when training them with R2-G2. We agree that the independence assumption is not necessary to derive the R2-G2 estimator and investigating the correlated noise setting serves as an interesting direction for future work! We note that the independence assumption affects model specification, which affects training with the RT estimator as much as it does the R2-G2 estimator.
>
> In our work, we have presented the R2-G2 estimator under the assumption of independent Gaussians to present the clear connection to Rao-Blackwellisation as it is one of our key contributions. While we agree that independent assumption is strong, we note that it is also a common assumption when evaluating a test bed of standard probabilistic models (e.g. VAEs, BNNs). We propose to state that the expression for the R2-G2 estimator presented in Definition 4.1 only holds under the independent Gaussian assumption and that a more general expression can be derived for the correlated Gaussian setting in a potential camera-ready version.
>
>
> >Moreover, it is unclear if $\epsilon^{\ast}$ refers to the mean of the distribution over $\epsilon$ as in Sec. 4.1 or not, as it is referred to as the "fitted noise" [...].
> > What is im(X) and ker(X)?
>
> [E] $\texttt{im}(\mathbf{X})$ and $\texttt{ker}(\mathbf{X})$ refer to the image and kernel of the matrix $\mathbf{X}$ respectively. We agree that introducing these quantities in Section 4.2 affects the readability of the section. In the next version, we propose to defer these quantities to the Appendix to improve readability of the paper.
>
> Indeed $\epsilon^{\ast}$ refers to both the mean of the conditional distribution $\epsilon| \mathbf{W} \cdot g = \mathbf{z}$ from Section 4.1 and the fitted noise from Line 178. Using the language of linear models, fitting a linear model (Equation after line 172) is the same as mapping “observed values” to “fitted values”. For consistency with this language, we propose to change Line 178 to “projecting the observed noise $\epsilon$ to the fitted noise $\epsilon^{\ast}$ [...]”.
>
> > What is the computational cost in FLOPS?
> > How does this approach scale and perform in more interesting settings (e.g., ImageNet)?
>
> [F] Great question! We first note that the worst-case complexity of computational costs from the conjugate gradient (CG) algorithm is $\mathcal{O}(m^3)$ (line 188) when one is attempting to invert a $m \times m$ dense matrix $C$ without knowing its structure. Here the worst-case would be running $m$ iterations of CG with each iteration requiring $m^2$ flops for a matrix-vector product.
>
> In the mean-field setting, we know more structure about the matrix $C$ and can factorise it as $C = \mathbf{W}\mathbf{V}\mathbf{W}^{\top}$ where $\mathbf{V} \in \mathbb{R}^{n \times n}$ is diagonal and $\mathbf{W} \in \mathbb{R}^{m \times n}$. Moreover, we know the rank of $V$ and the maximum rank of $\mathbf{W}$, so CG only needs to be run for $k=min(m,n)$ iterations at most, by using the property of ranks that $\text{rank}(AB) \leq min(\text{rank}(A), \text{rank}(B))$ for matrices $A,B$.
>
> A forward pass for a linear layer using the global reparameterisation trick would use a total of $(2m + 1)n$ flops for matrix-vector products. Using the R2-G2 estimator runs at most $k$ iterations of the CG algorithm with each iteration requiring $(2m + 1)n$ flops for matrix-vector products, so an additional $(2m + 1)nk$ flops at most.
>
> In our paper, we have used $n$ and $m$ to refer to the dimensions of the input and output/pre-activations Gaussian variables. For BNNs, we would have $n > m^2$ since there are more weights than pre-activations, so the cost complexity would be $\mathcal{O}(n)$ for a forward pass using the global reparameterisation trick and an additional $\mathcal{O}(n)$ for R2-G2. For VAEs, we would have $m > n$ since we map low-dimensional latents to a higher-dimensional space, so the cost complexity would be $\mathcal{O}(m)$ for a forward pass using the global reparameterisation trick and an additional $\mathcal{O}(mn)$ for R2-G2. In the next version, we propose to include this discussion in the Appendix.
>
> We note that using R2-G2 would be practical in the top/final layers of neural network architectures, even for larger architectures, for two reasons. First, these layers are where features are compressed to low-dimensions, usually with a linear layer, so this limits the number of iterations of conjugate gradient. Second, the variance of gradients of top/final layers are much higher than bottom/hidden layers (we can see this in the plots of RT gradient variances in Figures 1 and 3), so there is more benefit to Rao-Blackwellising gradients in top/final layers when training.
>
> > Why doesn't the convergence improve?
>
> [G] For BNNs, training with reduced-variance gradients (R2-G2) does not exhibit large gains over RT. We believe that gains in BNNs are more driven by a good architecture choice to the dataset, such as using a CNN for CIFAR-10. Please see the comments by Reviewer jPqY and Figure 1a of the original local reparameterisation trick paper [2] which shows similar error rates between the local and global reparameterisation tricks (referred to as Variational (A) and Variational (B) respectively in [2]).  We also note that the architectures have been simplified in our experiments, as we do not use dropout or batch normalisation layers. While this can have an effect on model performance, we wanted to better observe the variance reduction from R2-G2 in isolation (see Figures 1, 3, 4, 5) and whether it yields performance gains.
>
> For hierarchical VAEs, we show in Table 3 and Figure 2 that training the decoder improves the convergence of hierarchical VAE models. In the case of three-layer VAEs on MNIST and Omniglot datasets, we see an improvement of over 9 nats and 1 nat respectively which is substantial.
>
> * [1] Max B Paulus, Chris J. Maddison, and Andreas Krause. “Rao-Blackwellizing the Straight-Through Gumbel-Softmax Gradient Estimator”. In ICLR 2021.
> * [2] Durk P Kingma, Tim Salimans, and Max Welling. “Variational dropout and the local reparameterization trick”. In NeurIPS 2015.

---

> > ### Comment · Reviewer_ZPWE · 2025-08-01
> >
> > I want to thank the authors for their response; all of my questions have been sufficiently addressed, and I do not have any additional questions or concerns.

---

> > > ### Author Response · Authors · 2025-08-03
> > >
> > > We would like to thank the reviewer for their positive reception of our rebuttal response, and their valuable feedback which will allow us to provide an improved and clarified version of our submission in a potential camera-ready version. If all concerns have been addressed, we kindly request the reviewer to update their score.

---

> > > > ### Comment · Reviewer_ZPWE · 2025-08-04
> > > >
> > > > I have updated my score to an accept to reflect my positive opinion after the rebuttal.
> > > >
> > > > _Note, from what I understand from this year's process, authors might not be able to see the updated score before the final decision is made._

---

> > > > > ### Author Response · Authors · 2025-08-04
> > > > >
> > > > > We thank the reviewer for engaging with us during the discussion period, and further thank them for updating their score to reflect a positive reception of our work after the changes proposed in the rebuttal response. Thank you!

---

### Official Review · Reviewer_jPqY · 2025-07-02

**Clarity:** 4
**Significance:** 2
**Originality:** 3
**Rating:** 4
**Confidence:** 3

**Summary:**

This paper proposes the R2-G2 gradient estimator.
This gradient estimator generalizes the local reparameterization gradient by highlighting connections to Rao-Blackwellization and proposes an implementation for the general case of models with latent Gaussian variables.
The estimator is evaluated against the reparameterization gradient estimator in hierarchical variational auto-encoders and bayesian neural networks.

**Questions:**

- Can you expand discussion on the computational costs of the conjugate gradient method relative to the full forward and backward pass of the model?
- Can the authors clarify the backward pass? Does their implementation differentiate through the roll-outs conjugate gradient algorithm or otherwise?
- Can you provide results comparing R2-G2 and RT on the same compute budget to provide an equal playing field?

**Ethical Concerns:**

["NO or VERY MINOR ethics concerns only"]

**Final Justification:**

I read the rebuttal and my questions were clarified. The concern about the practical significance and the empirical improvements of this method remain. However, I still believe that the reasons to accept outweigh the reasons to reject, so I am keeping my score as is.

**Limitations:**

yes

**Quality:**

3

**Strengths And Weaknesses:**

Quality:
- The paper highlights some interesting connections between the local reparam trick and Rao-Blackwellization and R2-G2 naturally emerges as a generalization of the local reparam gradient as a result.
- The technical exposition in the paper is very clear, all claims are well-supported and results are derived.
- Kudos to the authors for being transparent and honest about the experimental results.
- Unfortunately, the method seems rather compute-intensive (running conjugage-gradient algorithm as part of the forward-pass), which makes me question how practically relevant it is.

Clarity
- This paper is extremely well-written and a pleasure to read.
- The experimental exposition given is thorough.
- The mathematical notation used is consistent and clear.

Significance
- I do believe that the theoretical analysis and insights into local reparam gradients, connections the paper draws to Rao-Blackwellization, and the generalized R2-G2 estimator the paper proposes are valuable to the research community.
- My main criticism is with the benefit of using the estimator in practise. The improvements in some cases are marginal (e.g., BNN on MNIST, CIFAR-10, no improvement over RT) or non-existent (VAE, one layer, as the authors acknowledge).  However, I appreciate the transparency and honesty about the experimental results. BNN on MNIST, CIFAR-10 which is equivalent to the method, also got published and doesn't do better in these settings than RT (as it's equivalent R2-G2). The variance reduction is clearly visible, but doesn't translate into improved performance, possibly because the tasks are too easy.  Further, the experimental evaluation is limited to "initial training". However, the authors make a good argument for motivating this setting, which I appreciate.

Originality
- In my opinion, the R2-G2 estimator provides a valuable new perspective on local reparm gradients. Rao-Blackwellization is of interest in the gradient estimation community too, so this work is a natural addition to existing work.

---

> ### Author Rebuttal · Authors · 2025-07-30
>
> We thank the reviewer for their review and appreciate their positive reception of our work. Please see our responses to each question raised individually below.
>
> ### Questions
>
> > Can you expand discussion on the computational costs of the conjugate gradient method relative to the full forward and backward pass of the model?
>
> [A] Great question! We first note that the worst-case complexity of computational costs from the conjugate gradient (CG) algorithm is $\mathcal{O}(m^3)$ (line 188) when one is attempting to invert a $m \times m$ dense matrix $C$ without knowing its structure. Here the worst-case would be running $m$ iterations of CG with each iteration requiring $m^2$ flops for a matrix-vector product.
>
> In the mean-field setting, we know more structure about the matrix $C$ and can factorise it as $C = \mathbf{W}\mathbf{V}\mathbf{W}^{\top}$ where $\mathbf{V} \in \mathbb{R}^{n \times n}$ is diagonal and $\mathbf{W} \in \mathbb{R}^{m \times n}$. Moreover, we know the rank of $V$ and the maximum rank of $\mathbf{W}$, so CG only needs to be run for $k=min(m,n)$ iterations at most, by using the property of ranks that $\text{rank}(AB) \leq min(\text{rank}(A), \text{rank}(B))$ for matrices $A,B$ of appropriate shapes.
>
> A forward pass for a linear layer using the global reparameterisation trick would use a total of $(2m + 1)n$ flops for matrix-vector products. Using the R2-G2 estimator runs at most $k$ iterations of the CG algorithm with each iteration requiring $(2m + 1)n$ flops for matrix-vector products, so an additional $(2m + 1)nk$ flops at most.
>
> In our paper, we have used $n$ and $m$ to refer to the dimensions of the input and output/pre-activations Gaussian variables. For BNNs, we would have $n > m^2$ since there are more weights than pre-activations, so the cost complexity would be $\mathcal{O}(n)$ for a forward pass using the global reparameterisation trick and an additional $\mathcal{O}(n)$ for R2-G2. For VAEs, we would have $m > n$ since we map low-dimensional latents to a higher-dimensional space, so the cost complexity would be $\mathcal{O}(m)$ for a forward pass using the global reparameterisation trick and an additional $\mathcal{O}(mn)$ for R2-G2. In the next version, we propose to include this discussion in the Appendix.
>
> > Can the authors clarify the backward pass? Does their implementation differentiate through the roll-outs conjugate gradient algorithm or otherwise?
>
> [B] Our implementation, presented as Algorithm 1, does not differentiate through $\beta^{\ast}$ or any of the roll-outs/iterates generated by the conjugate gradient algorithm. Once $\beta^{\ast}$  and $\epsilon^{\ast}$ are computed, they are only treated as constant vectors. Specifically, $\epsilon^{\ast}$ is used to compute $\mathbf{z}^{\ast} = \mathbf{W} \cdot (\mathbf{V} \odot \texttt{diag}(\epsilon^{\ast}))$ so that the Jacobian from Equation 8 within Definition 4.2 is used for backpropagation. Please also see our answer [A] to reviewer zzSq for a detailed explanation.
>
> > Can you provide results comparing R2-G2 and RT on the same compute budget to provide an equal playing field?
>
> [C] While a comparison can theoretically be made to compare R2-G2 and RT based on the number of FLOPs used for both estimators, we note that this is difficult to track in practice since we do not know the following beforehand: convergence behaviour of the conjugate gradient algorithm on the problem at hand and the exact FLOPs used before it terminates.
>
> In our hierarchical VAE comparisons, we fixed a small number of gradient steps (100K) and a limit in run-time on the same computing resource (a single NVIDIA V100 GPU). The run-time limit was set to 90 minutes for two-layer VAEs and 150 minutes for three-layer VAEs. Empirically, we observed that R2-G2 experiment runs were all completed in these limits with no more than 20 minutes of additional run-time compared to RT experiment runs. We propose to include these run-time limits and empirical observation within the Experiments section in a potential camera-ready version.

---

> > ### Comment · Reviewer_jPqY · 2025-08-06
> > **Thank you for your rebuttal!**
> >
> > I thank the authors for clarifying the points.
> > I believe that including the discussion about the computational costs is helpful.
> > Overall, my concerns about the practicality and empirical improvements of the method remain, but I still believe that the reasons for acceptance outweigh the reasons for rejection, and am keeping my score as is.

---

> > > ### Author Response · Authors · 2025-08-06
> > >
> > > We thank the reviewer for engaging with us during the discussion period, and further thank them for their kind comments vouching for the strengths in our work. We hope to address remaining concerns with support from relevant rebuttal responses to other reviewers.
> > >
> > > On the point of practicality, we have indeed showed that the R2-G2 estimator yields lower gradient variances in BNN linear and convolutional layers. Plots in Figures 1 and 3 also show that the variance of gradients in top/final layers usually much higher than bottom/hidden layers. This makes training with Rao-Blackwellised gradients practical and beneficial in top/final layers since this is where features are compressed to low-dimensions, even for larger architectures (please see our answer [B] to reviewer j5Ym).
> > >
> > > On the point of empirical improvements, we show that using the R2-G2 estimator improves the convergence of hierarchical VAEs. In the case of three-layer VAEs on MNIST and Omniglot datasets, we saw improvements of over 9 nats and 1 nat respectively which is substantial. For BNNs, training with reduced-variance gradients (R2-G2) does not exhibit large gains over RT. We believe that gains in BNNs are more driven by a good architecture choice to the dataset, such as using a CNN for CIFAR-10 (please see our answer [G] to reviewer ZPWE).
> > >
> > > We hope these address remaining concerns around practicality and improvements. Lastly, we thank the reviewer for maintaining a positive score and viewing to accept our work.

---

### Official Review · Reviewer_j5Ym · 2025-07-03

**Clarity:** 3
**Significance:** 3
**Originality:** 2
**Rating:** 4
**Confidence:** 4

**Summary:**

A common problem in probabilistic machine learning is to learn the parameters of a distribution q used to integrate out latent random variables from an objective function. In other words, the goal is to optimize the expectation of an objective function with respect to the distribution q by adjusting the parameters of q.  Because the expectation depends on q, the gradient cannot simply be pushed inside the expectation. For example, in Bayesian inference, the latent variables are the parameters of the model and the distribution q is the variational approximation to the posterior. In VAEs, the latent variables are the encodings and the distribution q is the encoder.

The authors start by discussing three common approximations, presented in decreasing order of variance: REINFORCE, which uses the log derivative trick, the reparameterization trick (RT), which requires q to be reparameterizable, and the local reparameterization trick (LRT), which applies when the latent variable can be expressed as an affine transformation of Gaussian random variables (for example, the pre-activations in a neural network). The local reparameterization trick tends to have the least variance, but also requires the latent variables to factorize under q, which happens, for example, with a mean-field Gaussian Bayesian neural network.

This paper introduces a new estimator, R2-G2, designed to acheive low variance like the local reparameterization trick but can apply when q does not factorize. The R2-G2 estimator performs Rao Blackwellization on the RT estimator by conditioning on a single sample of the pre-activations. As with any Rao Blackwellization, this means it will have lower variance. Note that the second in G R2-G2 is for Gaussian because their estimator applies to Gaussian q.

The paper then discusses computing their estimator by solving a least squares problem and shows that it is equivalent to the LRT for a mean-field Gaussian BNN.

There are two experiments in the paper: image classification with BNNs and hierarchical VAEs, both on standard benchmark datasets. For the BNN, the authors compare against RT and LRT on MNIST, and against RT on CIFAR-10. They do not compare against LRT on CIFAR-10 because they use convolutional layers, which the LRT does not support because q would not factorize. Looking at the results in Table 2, the confidence intervals appear to all overlap. For the VAE experiment, the authors compare against RT and find that R2-G2 has better ELBO value on test data.

**Questions:**

Questions:
- Could a non-mean-field example be run for the BNN, which would allow R2-G2 to be tested in a situation where the LRT doesn’t apply?
- Is the CNN example main architecture where R2-G2 applies but LRT does not? What about attention?
- Is there a reason why the LRT is not run for the VAE example? It is a fully factorized Gaussian variational posterior so my understanding is that it can be applied.
- Is computing $\beta^*$ practical for large architectures? How long did you need to run conjugate gradients? I did not see a description of the MLP architecture in the paper (the appendix references the main text and the main text cites Srivastava 2014, but this paper has many MLP examples).
Note the first 3 questions are about understanding where R2-G2 is most useful.

Small:
- MLP, ELBO (in intro) not defined
- The equation after line 63 seems to assume q can be reparameterized
- Notation: Is f parameterized by v?
- The equation after line 63 appears to assume $q$ is reparameterizable

**Ethical Concerns:**

["NO or VERY MINOR ethics concerns only"]

**Final Justification:**

Given that this method requires additional computation beyond RT, ideally there would be stronger empirical results, especially in the BNN example (as I interpret the results, all methods are about the same). However, the interpretation of LRT as Rao-Blackwellisation is a contribution on its own and there is stronger evidence that R2-G2 improves over RT in the VAE example where LRT does not apply (clarified to me during the rebuttal).

For what it is worth, the name “R2-G2” was not a factor in my review, but I did appreciate it.

**Limitations:**

Yes

**Paper Formatting Concerns:**

No concerns

**Quality:**

2

**Strengths And Weaknesses:**

The paper is clear and well written. The problem addressed by the paper is a generic and common problem in ML so any theoretical or practical improvements are significant.

Regarding originality, the paper is conceptually straightforward as it applies a well-known technique for variance reduction (Rao Blackwellisation) to a well-known estimator (RT). Proposition 4.2 seems like an expected result. Theorem 4.3 is more interesting to me because it provides an interpretation of LRT.

To me there are two questions the experiments should investigate: (1) is the estimator meaningfully reducing variance? (2) is the estimator computationally feasible when it is?

The latter question is not really addressed in the experiments since the architectures are small, I believe (though I am not sure what they are exactly, see question below).

Regarding first question, for the MNIST dataset the LRT and R2-G2 should be the same by Theorem 4.3 (which appears to be the case) but they have little to no improvement over RT. I would be interested in seeing a full covariance variational family (or something not factorized), since in this setting (as I understand it) R2-G2 is not guaranteed to be the same as LRT. Similarly for CNN on CIFAR-10, R2-G2 and RT have similar performance, but LRT doesn’t apply here.

For the VAE, I didn’t follow why LRT was not used (see mu question below). It is nice to see that R2-G2 had a larger improvement over RT than in the BNN case, but the confidence intervals were overlapping.

In general, my primary concern is that the experiments don’t seem to verify the issues of the LRT discussed in Section 4, namely performance like the LRT, better than RT, but in situations where it can’t be used (unless it can’t be used for the VAE).

---

> ### Author Rebuttal · Authors · 2025-07-30
>
> We thank the reviewer for their review and appreciate their engagement with our work. Please see our responses to each question raised individually below.
>
> ### Questions
>
> > Could a non-mean-field example be run for the BNN, which would allow R2-G2 to be tested in a situation where the LRT doesn’t apply?
>
> > Is the CNN example main architecture where R2-G2 applies but LRT does not? What about attention?
>
> > Is there a reason why the LRT is not run for the VAE example? It is a fully factorized Gaussian variational posterior so my understanding is that it can be applied.
>
> [A] Great questions! We note that the R2-G2 estimator is only applicable where the reparameterisation trick is used to sample Gaussian variables, we do not include attention layers since standard attention layers do not usually apply the reparameterisation trick to sample Gaussian variables. We also note that using the global reparameterisation trick on non-mean-field BNNs would introduce a separate scalability challenge that would be difficult to tackle within this work. To see this, we can take a BNN linear layer has a $m \times n$ matrix of Gaussian weights as an example. This would require parameterising a dense $mn \times mn$ covariance matrix (or its Cholesky factor as a lower triangular matrix), making computations challenging for both the reparameterisation trick and the R2-G2 estimator by extension.
>
> Mean-field BNN convolutional layers and VAEs are both examples where LRT cannot be applied. To clarify the VAE example, the fully factorised Gaussian assumption is applied in the latent space $\mathbf{v}$ so the reparameterisation trick applied here is the global reparameterisation trick and RT estimator. That is, the local reparameterisation trick and LRT estimator are not usable for VAEs because they work by summing over multiple independent Gaussian variables that are mapped to scalar Gaussian pre-activations. The linear transformation of these VAE latents $\mathbf{W}\mathbf{v}$ (i.e. pre-activations) does not have a factorised Gaussian variational posterior in general, so LRT cannot be applied. We make this more precise below.
>
> Let $\mathbf{v} \in \mathbb{R}^n$ be a vector of zero-mean independent Gaussian variables with variance parameters $(\sigma^{(1)})^2, \dots, (\sigma^{(n)})^2$ contained in a diagonal covariance matrix $\mathbf{V}$. When $\mathbf{v}$ is mapped to a scalar output $z = \mathbf{w}^{\top}\mathbf{v} \in \mathbb{R}$ for a vector $\mathbf{w} \in \mathbb{R}^n$, the covariance matrix of $z$ is just a positive scalar $\sigma_{z}^2 = \sum_{i=1}^{n} w_i^2 (\sigma^{(i)})^2$. The local reparameterisation trick exploits this convenience of working with scalars to create $m$ scalar pre-activations that are concatenated and output as a vector of Gaussian pre-activations $\mathbf{z} \in \mathbb{R}^m$, by working with $m$ Gaussian vectors $\mathbf{v}^{(1)},\dots ,\mathbf{v}^{(m)} \in \mathbb{R}^n$. The key point is that the covariance matrix of $\mathbf{z}$ is a diagonal matrix by design to make sampling scalar pre-activations easy in the forward pass.
>
> The subtle point here on local reparameterisation is that by square-rooting the pre-activation variance $\sigma_{z}^2$, it actually configures reduced-variance gradients in the backward pass. The intuition here is that the square-root function $y=x^{1/2}$ has a derivative which looks like its inverse as $dy/dx = 1/2 \times x^{-1/2}$, so applying the local reparameterisation trick in the forward pass is actually inverting the (scalar) covariance matrix $\sigma_{z}^2$ in the backward pass. The purpose of Theorem 4.3 is to formalise this observation as Rao-Blackwellised gradients and is therefore a special case of the R2-G2 estimator.
>
> We note that using a diagonal covariance matrix for Gaussian pre-activations only works for BNN linear layers in the independent Gaussian weights (i.e. mean-field) setting. In general, we would use a matrix $\mathbf{W} \in \mathbb{R}^{m \times n}$ to output a pre-activation vector $\mathbf{z} = \mathbf{W}\mathbf{v}$. Here $\mathbf{z}$ has covariance matrix $\mathbf{W}\mathbf{V}\mathbf{W}^{\top}$ which is not diagonal, so LRT cannot be applied in such cases but R2-G2 can be applied. A notable example is for hierarchical VAE architectures where $\mathbf{v}$ is a vector of latent Gaussian variables used in the decoder and $\mathbf{W}$ is the linear transformation that immediately follows it (please see our VAE experiments in Section 5.3).
>
> To summarise, LRT and R2-G2 estimators both invert the covariance matrix of Gaussian pre-activations to produce reduced-variance gradients. LRT inverts a (scalar) matrix automatically as a result of the parameterisation used in the forward pass, but the application is restricted to BNN linear layers. On the other hand, R2-G2 inverts a symmetric square matrix after observing pre-activations in the forward pass, by using the conjugate gradient algorithm as a matrix inversion procedure and thereby making it more general than LRT. To further motivate the R2-G2 estimator, we propose to include this explanation at the start of Section 4 in a potential camera-ready version. We agree that an even more general non-mean-field setting (where $\mathbf{V}$ is not diagonal) will be a very interesting avenue for future work and further experiments!
>
> > Is computing $\beta^{\ast}$ practical for large architectures? How long did you need to run conjugate gradients? I did not see a description of the MLP architecture in the paper (the appendix references the main text and the main text cites Srivastava 2014, but this paper has many MLP examples).
>
> [B] The computation of $\beta^{\ast}$ is done for each layer where the R2-G2 estimator is applied. This would be practical in the top/final layers of neural network architectures, even for larger architectures, for two reasons. First, these layers are where features are compressed to low-dimensions, usually with a linear layer, so this should limit the number of iterations of conjugate gradients. Second, the variance of gradients of top/final layers are usually much higher than bottom/hidden layers (we can see this in the plots of RT gradient variances in Figures 1 and 3), so there is more benefit to Rao-Blackwellising gradients in top/final layers when training.
>
> In our experiments, we ran conjugate gradient (CG) using the property of ranks that $\text{rank}(AB) \leq min(\text{rank}(A), \text{rank}(B))$ for matrices $A,B$ of appropriate shapes. For each NN layer where the R2-G2 is used, this equates to running maximum of: 1 CG iteration for mean-field BNN linear layers, 4 to 16 CG iterations for mean-field BNN convolutional layers, 50 CG iterations for VAE sampling layers in the decoder.
>
> The MLP architecture we used from Srivastava 2014 was a MLP with two hidden layers of 1024 ReLU units. We propose to explicitly state this detail in Section 5.2 of the Experiments section in a potential camera-ready version.
>
> Small questions
> > MLP, ELBO (in intro) not defined
> > The equation after line 63 seems to assume q can be reparameterized
> > Notation: Is f parameterized by v?
> > The equation after line 63 appears to assume is reparameterizable
>
> [C] Thanks for these comments as it will improve clarity of our writing. In the next version, we propose to introduce MLP within the Experiments section and ELBO in Line 57, and to write $\ell_{\mathcal{D}, f}(\mathbf{v})$ in the Equation after Line 51 and $\ell(\mathbf{v})$ in the Equation after Line 63 to describe the problem setting without assuming $q$ is reparameterisable in Section 2.

---

> ### Comment · Reviewer_j5Ym · 2025-08-04
>
> I appreciate the detailed rebuttal by the authors and have increased my score as a result. The explanation of why the LRT does not apply to VAEs was helpful. I don’t completely agree regarding the scalability challenges of a non-mean-field family for a BNN layer. It could be a low rank family (e.g, as in this paper https://proceedings.neurips.cc/paper/2020/file/310cc7ca5a76a446f85c1a0d641ba96d-Paper.pdf) and, as the authors point out, R2-G2 would only be practical in the last few layers anyway (where the dimension might not be too big).
>
> One thought that might make the paper stronger: if you could demonstrate on the VAE example that R2-G2 is worth the additional computation. For example, what happens if you increase the number of ELBO samples for RT so that the run time is about the same as R2-G2. Does R2-G2 still outperform? (Not expecting an answer)

---

> > ### Author Response · Authors · 2025-08-05
> >
> > We thank the reviewer for engaging with us during the discussion period, and further thank them for increasing their score after our rebuttal response. Indeed, we agree that a low-rank family is a natural extension of BNN layers tested in our work. While challenging to include in this work, as it would affect presentation of the R2-G2 estimator and its connection to the LRT estimator as our contribution (please see our answer [D] to reviewer ZPWE), it sets an interesting direction for future work in the correlated Gaussian setting. Thank you!

---

### Official Review · Reviewer_zzSq · 2025-07-03

**Clarity:** 3
**Significance:** 3
**Originality:** 3
**Rating:** 5
**Confidence:** 3

**Summary:**

The paper considers the following problem: estimate the gradient of an expectation that involves linear transformations of Gaussian random variables, in an unbiased and low-variance way. They propose the R2-G2 estimator, which Rao-Blackwellizes the reparameterization gradient estimator by conditioning on the linear transformation results of Gaussian variables. They prove that their estimator is unbiased and has lower variance than the reparameterization estimator. They provide closed-forms of their estimator and design an efficient (approximate) method for computing the estimator. They further prove that an existing estimator (called the local reparameterization gradient estimator) is a special case of their estimator. They empirically demonstrate that their estimator outperforms the reparameterization estimator.

**Questions:**

## Questions
- I could not understand why running Algorithm 1 (and performing a backward pass) approximately computes the R2-G2 estimator as defined in Definition 4.1. More concretely, I understood the text in Section 4.2 before the paragraph "Iterative solver", but could not figure out why differentiating $z^*$ yields the R2-G2 estimator. Could you explain it in more detail?

- Section 4.3 establishes that the local reparameterization estimator (LRT) is a special case of the R2-G2 estimator. However, I could not fully understand how much more general the R2-G2 estimator is than the LRT. Could you give concrete (preferably simple) examples where the R2-G2 is applicable but the LRT is not?

## Comments
- Section 3: The loss function $\ell$ in Eq (1) depends on $v$ as well as $\theta$, and this dependency looks required to express the ELBO (lines 58--59). However, in Section 3, the equations of existing estimators (SCORE, Reparameterization trick, ...) do not contain the $\nabla_\theta \ell$ term. I guess we should either include the term there, or make an additional assumption that $\ell$ does not directly depend on $\theta$?

**Ethical Concerns:**

["NO or VERY MINOR ethics concerns only"]

**Final Justification:**

I increased my score from 4 to 5 because the rebuttal addressed my main concerns.

**Limitations:**

Yes, some limitations are discussed in Section 6.

**Paper Formatting Concerns:**

None.

**Quality:**

3

**Strengths And Weaknesses:**

## Strengths
- The paper tackles an important problem: develop an unbiased, low-variance estimator of the gradient of an expectation.
- The paper proposes a novel method for the problem. Their method generalizes an existing method (local reparameterization gradient) and outperforms a popular method (global reparameterization gradient).
- The paper is backed by both theoretical and empirical results. Theoretically, they prove the unbiasedness and a low-variance property of their method, and show a formal result that explains why an existing method (local reparameterization gradient) has performed well in practice; empirically, they demonstrate these claims in practical settings.
- The paper is mostly well-written and well-structured.

## Weaknesses
- It was not so clear to me how their main algorithm (Algorithm 1) implements their method (Definition 4.1).
- I did not fully understand the degree to which their method generalizes an existing method (local reparameterization gradient).

---

> ### Author Rebuttal · Authors · 2025-07-30
>
> We thank the reviewer for their review and appreciate their positive reception of our work. Please see our responses to each point raised individually below.
>
> ### Weaknesses/Questions
> > It was not so clear to me how their main algorithm (Algorithm 1) implements their method (Definition 4.1).
>
> [A] We note that a forward pass consisting of reparameterisation with diagonal $\mathbf{V} \in \mathbb{R}^{n \times n}$ and linear transformation $\mathbf{W} \in\mathbb{R}^{m \times n}$ will output
> $\mathbf{z} = \mathbf{W} \cdot (\mathbf{V} \odot \texttt{diag}(\epsilon))$.
>
> This gives us the gradient
> $J_{\mathbf{z}}(\theta) = \left(\mathbf{W} \cdot
>     \begin{bmatrix}
>     \begin{array}{c|c}
>          \mathbf{I}_n & \frac{1}{2}\Sigma^{-\frac{1}{2}} \odot (\mathbf{1}_n \epsilon^{\top})
>     \end{array}
>     \end{bmatrix}
>     \right)^{\top}
> $
> in the backward pass.
>
> By swapping $\epsilon$ with $\epsilon^{\ast}$, we have a forward pass outputting $\mathbf{z}^{\ast} = \mathbf{W} \cdot (\mathbf{V} \odot \texttt{diag}(\epsilon^{\ast}))$ with gradient
> $J_{\mathbf{z}^{\ast}}(\theta) = \left(\mathbf{W} \cdot
>     \begin{bmatrix}
>     \begin{array}{c|c}
>          \mathbf{I}_n & \frac{1}{2}\Sigma^{-\frac{1}{2}} \odot (\mathbf{1}_n (\epsilon^{\ast})^{\top})
>     \end{array}
>     \end{bmatrix}
>     \right)^{\top}
> $
>
> Here $J_{\mathbf{z}^{\ast}}(\theta)$ is the conditional expectation of  $J_{\mathbf{z}}(\theta)$, using the conditional distribution $\epsilon| \mathbf{z}$ with mean $\epsilon^{\ast}$. Algorithm 1 solves for $\epsilon^{\ast}$ with conjugate_gradient and returns stop_gradient$(\mathbf{z}-\mathbf{z}^{\ast}) + \mathbf{z}^{\ast}$. The return step outputs $\mathbf{z}$ in the forward pass but uses $J_{\mathbf{z}^{\ast}}(\theta)$ instead of $J_{\mathbf{z}}(\theta)$ in the backward pass, thereby implementing Equation 8 from Definition 4.1.
>
>
> > Section 4.3 establishes that the local reparameterization estimator (LRT) is a special case of the R2-G2 estimator. However, I could not fully understand how much more general the R2-G2 estimator is than the LRT. Could you give concrete (preferably simple) examples where the R2-G2 is applicable but the LRT is not?
>
> [B] LRT and R2-G2 estimators both use the known mean of conditional Gaussian distributions to produce Rao-Blackwellised gradients, by inverting the covariance matrix of pre-activations. As an analogy, the R2-G2 estimator generalises the LRT estimator in the same manner as inversion of a square matrix generalises inverting a scalar. We make this point more concrete below.
>
> Let $\mathbf{v} \in \mathbb{R}^n$ be a vector of zero-mean independent Gaussian variables with variance parameters $(\sigma^{(1)})^2, \dots, (\sigma^{(n)})^2$ contained in a diagonal covariance matrix $\mathbf{V}$. When $\mathbf{v}$ is mapped to a scalar output $z = \mathbf{w}^{\top}\mathbf{v} \in \mathbb{R}$ for a vector $\mathbf{w} \in \mathbb{R}^n$, the covariance matrix of $z$ is just a positive scalar $\sigma_{z}^2 = \sum_{i=1}^{n} w_i^2 (\sigma^{(i)})^2$. The local reparameterisation trick exploits this convenience of working with scalars to create $m$ scalar pre-activations that are concatenated and output as a vector of Gaussian pre-activations $\mathbf{z} \in \mathbb{R}^m$, by working with $m$ Gaussian vectors $\mathbf{v}^{(1)},\dots ,\mathbf{v}^{(m)} \in \mathbb{R}^n$. The key point is that the covariance matrix of $\mathbf{z}$ is a diagonal matrix by design to make sampling scalar pre-activations easy in the forward pass.
>
> The subtle point here on local reparameterisation is that by square-rooting the pre-activation variance $\sigma_{z}^2$, it actually configures reduced-variance gradients in the backward pass. The intuition here is that the square-root function $y=x^{1/2}$ has a derivative which looks like its inverse as $dy/dx = 1/2 \times x^{-1/2}$, so applying the local reparameterisation trick in the forward pass is actually inverting the (scalar) covariance matrix $\sigma_{z}^2$ in the backward pass. The purpose of Theorem 4.3 is to formalise this observation as Rao-Blackwellised gradients and is therefore a special case of the R2-G2 estimator.
>
> On the point of generality of R2-G2, we note that using a diagonal covariance matrix for Gaussian pre-activations only works for BNN linear layers in the independent Gaussian weights (i.e. mean-field) setting. In general, we would use a matrix $\mathbf{W} \in \mathbb{R}^{m \times n}$ to output a pre-activation vector $\mathbf{z} = \mathbf{W}\mathbf{v}$. Here $\mathbf{z}$ has covariance matrix $\mathbf{W}\mathbf{V}\mathbf{W}^{\top}$ which is not diagonal, so LRT cannot be applied in such cases but R2-G2 can be applied. A notable example is for hierarchical VAE architectures where $\mathbf{v}$ is a vector of latent Gaussian variables used in the decoder and $\mathbf{W}$ is the linear transformation that immediately follows it (please see our VAE experiments in Section 5.3).
>
> To summarise, LRT and R2-G2 estimators both invert the covariance matrix of Gaussian pre-activations to produce reduced-variance gradients. LRT inverts a (scalar) matrix automatically as a result of the parameterisation used in the forward pass, but the application is restricted to BNN linear layers. On the other hand, R2-G2 inverts a symmetric square matrix after observing pre-activations in the forward pass, by using the conjugate gradient algorithm as a matrix inversion procedure and thereby making it more general than LRT. To further motivate the R2-G2 estimator, we propose to include this explanation at the start of Section 4 in a potential camera-ready version.
>
> > Section 3: The loss function $\ell$ in Eq (1) depends on $\mathbf{v}$ as well as $\theta$, and this dependency looks required to express the ELBO (lines 58--59). However, in Section 3, the equations of existing estimators (SCORE, Reparameterization trick, ...) do not contain the term $\nabla_{\theta} \ell$. I guess we should either include the term there, or make an additional assumption that does not directly depend on $\theta$?
>
> [C] Thank you for spotting the typo in the ELBO (lines 58--59). We agree that the dependency on $\theta$ can be better clarified here. In the next version, we propose to write $\ell_{\mathcal{D}, f}(\mathbf{v})$ in the Equation after Line 51 and $\ell(\mathbf{v})$ in the Equation after Line 63. This would better describe the problem setting without assuming $q$ is reparameterisable in Section 2. We also propose to only make an explicit dependency on $\theta$ in Section 3 when $q$ is assumed to be reparameterisable. Please also see our answer [C] to reviewer j5Ym.
>
> To clarify the equations in Section 3, we note that the gradient term $\nabla_{\theta} \ell$ is generally unknown and is exactly why we need gradient estimators! We have followed the hat notation from statistics to note that equations in Section 3 are estimators of $\nabla_{\theta} \ell$.

---

> > ### Comment · Reviewer_zzSq · 2025-08-05
> >
> > I thank the authors for their responses, which have addressed my concerns satisfactorily. I will increase my score.
> >
> > I hope that in the next version of the paper, the authors will include (i) the intuitive difference between LRT and R2-G2, and (ii) an explanation of how Algorithm 1 connects to Definition 4.1. Regarding (ii), I suggest including the specification of `conjugate_gradient(A, z)` (which I believe returns the solution to $A A^T x = z$, not $A x = z$).

---

> > > ### Author Response · Authors · 2025-08-05
> > >
> > > We thank the reviewer for engaging with us during the discussion period, and further thank them for increasing their score after the changes proposed in the rebuttal response. We agree that (i) and (ii) make great additions that better present the motivation and implementation of the R2-G2 estimator in the next version. Thank you!

---

### Author Response · Authors · 2025-08-04

We thank all reviewers for assessing our submission once more. With the discussion period ending soon, we would like to confirm with reviewers whether our responses have addressed concerns around our work.

---

### Decision · Program_Chairs · 2025-09-17

**Decision:**

Accept (poster)

**Comment:**

The paper proposes a reparameterization gradient for Monte Carlo gradient estimation with lower variance compared to existing approaches. The paper is clearly written, the approach is novel, and the method is supported by strong numerical experiments. That said, there are some limitations to the method such as higher computational cost, and the contribution is on the more incremental side.